# Cross-Layer Discrete Concept Discovery for Interpreting Language Models

## Abstract

Interpreting language models remains challenging due to the existence of residual stream, which linearly mixes and duplicates information across adjacent layers. This leads to the under-detection of features that exist in the specific layer being analyzed. Current research works either analyze neural representations at single layers, thereby overlooking this cross-layer superposition, or utilize a cross-layer variant of sparse autoencoder (SAE) for analysis. However, SAEs operate in continuous space, so there are no clear boundaries between neurons representing different concepts. We address these limitations by introducing Cross-Layer vector quantized-variational autoencoder (VQ-VAE), a novel framework that maps representations across layers through vector quantization. This causes the collapse of duplicated features in the residual stream, thus resulting in compact, interpretable concept vectors. Our approach combines top-$k$ temperature-based sampling during quantization with exponential moving average (EMA) codebook updates, providing controlled exploration of the discrete latent space while maintaining codebook diversity. Our experiments show that this framework, when combined with appropriate initialization, can effectively discover meaningful concepts. Our quantitative and qualitative experiments on the ERASER-Movie, Jigsaw, and AGNews datasets show that cross-layer VQ-VAE (CLVQ-VAE) can discover meaningful concepts that explain model predictions. [1]

## 1  Introduction

Large language models (LLMs) have demonstrated remarkable capabilities across a wide range of natural language processing tasks, yet their internal mechanisms remain largely opaque. This opacity poses major challenges for interpretability, limiting our scientific understanding and raising concerns around trust, accountability, and responsible use (Dodge et al., 2021; Sheng et al., 2021).

The majority of interpretability research focus on representations at individual layers – either by analysis of the activation patterns of neurons (Zhang et al., 2021) or by using probing classifiers that map the hidden states into pre-defined concepts (Belinkov et al., 2017; Arps et al., 2022; Kumar et al., 2023). These single-layer methods fail to account for how transformer residual streams duplicate and mix information across layers, meaning the computational structure that only becomes visible when examining multiple layers together, gets obscured (Team, 2024).

Recent advances in SAE methods (Härle et al., 2024; Lan et al., 2025) and transcoder architectures (Marks et al., 2024; Dunefsky et al., 2024a) have highlighted the value of analyzing layer pairs. Empirical studies show that cross-layer analysis often results in more interpretable features than single-layer approaches (Shi et al., 2025; Balagansky et al., 2025; Laptev et al., 2025). This is largely due to the additive residual stream in transformers, i.e., each layer contributes to the running representation, which causes features to persist and appear duplicated when layers are viewed in isolation (Lindsey et al., 2025). However, SAE-based methods operate in continuous spaces where concepts "split" across SAE neurons (Bricken et al., 2023). This splitting complicates identifying the linear combination of decoder vectors required to reconstitute the concept, hence forcing the use of arbitrary thresholds to isolate sparse activations (Oozeer et al., 2025).

---

[1]Anonymized code repository can be found here: https://anonymous.4open.science/r/CLVQVAE-09E9

This continuity limits interpretability because the resulting individual neurons don't align with the discrete conceptual categories humans use for reasoning (Wu et al., 2024).

VQ-VAEs have been extensively explored in computer vision to discretize the continuous representations of images into codebook vectors (van den Oord et al., 2018; Takida et al., 2022; Razavi et al., 2019). We hypothesize that when VQ-VAEs are applied to language model activations, the codebook vectors will capture interpretable linguistic concepts – such as syntactic patterns or semantic categories – that are essential for reconstruction objective of VQ-VAE. Also, even though VQ-VAEs share the reconstruction objective of SAEs, they critically differ by utilizing a single discrete codebook vector rather than a linear combination of active decoder vectors. This discrete bottleneck naturally concentrates information, effectively sidestepping the ambiguity of identifying which vectors to combine.

Building on these insights, we propose CLVQ-VAE, a framework that discovers concepts across transformer layers. Unlike standard VQ-VAEs that reconstruct the same layer, our model acts as a transcoder, i.e., mapping activations from a lower layer $l$ to a higher layer $h$ through a discrete bottleneck, thus collapsing redundant residual-stream features into interpretable codebook vectors. We further improve this architecture by introducing a stochastic sampling mechanism that selects from the top-$k$ nearest codebook vectors using temperature-controlled probability distributions, resulting in better codebook utilization and concept diversity compared to deterministic approaches.

We evaluate CLVQ-VAE on the ERASER-Movie (Pang & Lee, 2004), Jigsaw Toxicity (cjadams et al., 2017), and AGNEWS (Gulli, 2005) datasets using fine-tuned RoBERTa (Liu et al., 2019b), BERT (Devlin et al., 2019), and decoder-only models like LLaMA-2-7b and Qwen2.5-3B-Instruct. Perturbation-based experiments shows that our approach identifies salient concepts that strongly influence the predictions, outperforming clustering, single-layer VQVAE, and SAE baselines. The quality of these concepts is further validated by an LLM-as-a-judge evaluation, which finds them more coherent than those from the competing methods. Finally, human evaluation confirms their practical utility for interpretation, with CLVQ-VAE visualizations achieving higher model-alignment and inter-annotator agreement scores than the clustering baseline.

## 2 Methodology

We propose CLVQ-VAE, a modular framework that discovers concepts by reconstructing higher-layer activations using quantized representations of the lower-layer activations. As shown in Figure 1, the framework processes layer activations through three core components:

1. **Adaptive Residual Encoder:** This component applies controllable interpolation to input embeddings from a lower layer, hence preserving the semantic information encoded in the pre-trained representations.

2. **Vector Quantizer:** This acts as a discrete bottleneck, which maps the continuous encoder outputs to one of the codebook vectors, forcing the model to represent information compactly.

3. **Transformer Decoder:** Lastly, this component takes the codebook vectors corresponding to the encoder output and reconstructs the target activations from a higher layer, hence learning to predict the model's cross-layer computations from the discrete concepts alone.

These components are jointly optimized through a reconstruction loss that encourages the encoder to map inputs to relevant codebook vectors while training the decoder to accurately reconstruct higher-layer activations from these quantized representations.

### 2.1 Problem Formulation

We formalize concepts as vectors in a learned codebook $\mathcal{E} = \{\mathbf{e}_j\}_{j=1}^{K}$, where $\mathbf{e}_j \in \mathbb{R}^d$ and each corresponds to a cluster of semantically related token representations. Unlike continuous sparse autoencoder activations, our approach enforces discrete assignments: each encoder output $\mathbf{z}_e$ from layer $l$ maps to the closest codebook vector $\mathbf{z}_q \in \mathcal{E}$, where closeness is measured by Euclidean distance.

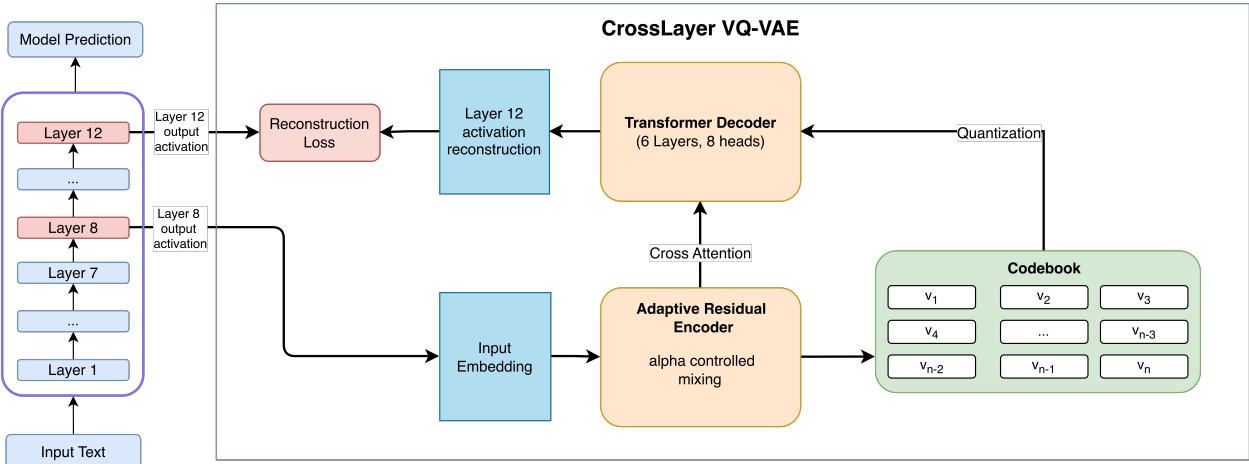

Figure 1: Overview of the CLVQ-VAE framework for cross-layer concept discovery. Lower-layer activations are passed through an adaptive residual encoder, discretized via vector quantization into concept vectors, and decoded to reconstruct higher-layer representations.

## 2.2 Adaptive Residual Encoder

Transformer-based language models produce rich, contextualized embeddings at each layer that have been shown to encode diverse linguistic information (Liu et al., 2019a; Sajjad et al., 2022b). A major challenge in adapting the VQ-VAE encoder to language model embeddings lies in determining the appropriate amount of embedding changes. These changes must be sufficient to enable codebook reassignment for effective reconstruction, yet constrained enough to preserve the semantic knowledge encoded in the embeddings.

To address this, we propose an adaptive residual encoder that implements controlled manipulation of the input embeddings. Rather than completely transforming the input, which would destroy valuable linguistic features due to random initialization of encoder parameters, or leaving it unchanged, which would limit concept discovery, our encoder introduces a learnable interpolation mechanism that respects the information-rich nature of embeddings while enabling targeted refinements for cross-layer reconstruction.

Given an input embedding $\mathbf{x} \in \mathbb{R}^d$ from layer $l$, the encoder produces an output $\mathbf{z}_e$ through following:

$$\mathbf{z}_e = (1 - \alpha) \cdot \mathbf{x} + \alpha \cdot \text{LN}(W\mathbf{x} + \mathbf{b}) \tag{1}$$

where $\alpha = \sigma(a) * 0.5$ is a mixing coefficient constrained to $[0, 0.5]$ with $a$ being a learnable parameter, and LN is the layer normalization applied to the linearly transformed $x$. We limit $\alpha$ to a maximum of 0.5 to prevent excessive modification of the original embedding, as we empirically found in Table 12 that allowing complete transformation ($\alpha \in [0,1]$) resulted in reduced codebook utilization.

In Table 12, we also observe that for an adaptive $\alpha$, the model initially prefers a small value of $\alpha$, which constrains the modification of the original embedding. As training progresses and encoder parameters get trained, the gradients gradually increase $\alpha$, allowing the model to make progressively larger modifications.

## 2.3 Vector Quantizer

The vector quantizer maps the encoder outputs to discrete concept representations. To promote the stable training and effective codebook utilization, we utilize three key mechanisms, i.e., k-means-based codebook initialization, temperature-controlled top-k sampling and the EMA-based codebook updates.

### 2.3.1 Codebook Initialization

The initial state of the codebook can significantly impact training stability and the quality of the learned concepts. We explore two main initialization approaches: **random initialization** and **k-means based initialization**, which itself has two variants. Each offers a different trade-off between simplicity, computational cost, and alignment with the data's underlying structure.

**Random Initialization:** A naive baseline where we randomly sample $K$ unique embedding vectors from the training dataset to serve as the initial codebook entries. While computationally inexpensive, this approach does not guarantee that the initial codebook vectors are representative of the overall data distribution, which can lead to slower convergence or "dead" codes.

**K-Means Initialization:** A more informed approach is to initialize the codebook with the centroids of clusters found in the input data. This method provides better initial coverage of the data distribution compared to random sampling and often leads to faster, more stable training. We evaluate two variants:

1. **Default K-Means.** Widely adopted in the VQ-VAE literature Łańcucki et al. (2020); Huh et al. (2023); Zeghidour et al. (2021), this variant applies the standard k-means algorithm to the entire set of training embeddings from layer $l$. The resulting $K$ centroids, which represent the means of the identified clusters in Euclidean space, are then used as the initial codebook vectors $\mathbf{e}_j$.

2. **Spherical K-Means.** This variant clusters vectors by angular similarity (cosine distance) rather than Euclidean distance, motivated by the observation that semantic similarity in NLP is often represented by vector direction (Banerjee et al., 2005). Input embeddings $\mathbf{x}_i$ are first unit-normalized ($\hat{\mathbf{x}}_i = \frac{\mathbf{x}_i}{\|\mathbf{x}_i\|_2}$), then clustered using standard k-means on the hypersphere. The resulting unit-vector centroids $\mathbf{c}_j$ are rescaled by the average magnitude of their assigned vectors to reintroduce magnitude information:

$$\mathbf{e}_j = \mathbf{c}_j \cdot \frac{1}{|C_j|} \sum_{i \in C_j} \|\mathbf{x}_i\|_2 \tag{2}$$

where $C_j$ is the set of original vectors assigned to cluster $j$. This initialization results in codebook vectors that group semantically similar words together while preserving magnitude information.

### 2.3.2 Top-k Temperature-Based Codebook Sampling

Our vector quantization mechanism employs temperature-based sampling (Takida et al., 2022) from the top-$k$ nearest codebook vectors. For each encoder output $\mathbf{z}_e$, we compute distances to all codebook vectors $\{\mathbf{e}_j\}_{j=1}^K$ as:

$$d(\mathbf{z}_e, \mathbf{e}_j) = \|\mathbf{z}_e - \mathbf{e}_j\|_2^2 \tag{3}$$

Rather than deterministically selecting the closest vector, we identify the top-$k$ nearest codebook vectors and sample from them using a temperature-controlled distribution:

$$p(j|\mathbf{z}_e) = \frac{\exp(-d(\mathbf{z}_e, \mathbf{e}_j)/\tau)}{\sum_{j' \in \text{top-}k} \exp(-d(\mathbf{z}_e, \mathbf{e}_{j'})/\tau)} \tag{4}$$

where $\tau$ is the temperature parameter. In our optimal configuration, we set $k = 5$ and $\tau = 1.0$, balancing exploration with exploitation. This controlled stochasticity during training encourages more uniform codebook utilization, reduces codebook collapse, and improves concept diversity (Appendix D.6).

### 2.3.3 EMA-Based Codebook Updates

To ensure the stable training dynamics, codebook is updated using the EMA (Łukasz Kaiser et al., 2018) rather than the direct backpropagation. The gradient-based updates can oscillate or become unstable with

the discrete assignments (van den Oord et al., 2018), and thus we instead maintain the running averages of both the assignment counts and the accumulated vectors for each codebook entry.

For each codebook vector $\mathbf{e}_j$ in a training batch, we update its accumulated vector sum $\mathbf{m}_j$ and its total assignment count $N_j$ with a decay factor $\gamma = 0.99$. The codebook vector is then updated to the mean of its assigned embeddings by normalizing the accumulated sum by the count. This update strategy, when combined with our stochastic top-$k$ sampling, ensures the codebook remains diverse and active throughout the training while converging toward the stable representations of the cross-layer transformations. The complete update equations are provided in Appendix B.1.

## 2.4 Transformer Decoder

The final component of our CLVQ-VAE architecture is a transformer-based decoder that maps the sequence of the quantized representations to the higher-layer activations. It consists of 6 layers with 8 attention heads each and uses both the self-attention and the cross-attention mechanisms.

Because the decoder reconstructs the target activations for the entire input sequence at once rather than generating them sequentially, a causal mask is unnecessary. The self-attention is therefore fully bidirectional, allowing each token to draw the context from the entire sequence to build a more accurate representation.

The cross-attention mechanism allows the decoder to leverage information from the unquantized encoder outputs. Similar to the residual connection in skip-transcoders (Dunefsky et al., 2024b), which improves reconstruction without compromising interpretability, the cross-attention mechanism offloads low-level reconstruction details, enabling the codebook to focus on distinct, high-level concepts as demonstrated in auxiliary bottleneck models (Sheth & Kahou, 2023). We provide empirical evidence in Appendix D.3 that this cross-attention mechanism indeed improves interpretability and reconstruction.

## 2.5 Training Objectives

Our training incorporates two weighted objectives combined into a single loss function. The primary objective is the reconstruction loss, which minimizes the mean squared error between decoder output $\hat{\mathbf{y}}$ and target higher-layer representation $\mathbf{y}$, defined as $\mathcal{L}_{\text{rec}} = \|\mathbf{y} - \hat{\mathbf{y}}\|_2^2$ (Dunefsky et al., 2024a). This ensures that the model captures the transformations occurring between neural network layers. Additionally, we employ a commitment loss that encourages encoder outputs to commit to codebook vectors, calculated as $\mathcal{L}_{\text{commit}} = \|\mathbf{z}_e - \text{sg}(\mathbf{z}_q)\|_2^2$, where sg denotes the stop-gradient operator. The total loss is given by $\mathcal{L}_{\text{total}} = \mathcal{L}_{\text{rec}} + \beta \mathcal{L}_{\text{commit}}$, where $\beta$ controls the relative importance of the commitment term. We set $\beta = 0.1$ (van den Oord et al., 2018) to avoid constraining the encoder output too strictly (Wu & Flierl, 2019).

# 3 Experimental Setup

**Data**   We conducted experiments for CLVQ-VAE on three datasets: ERASER-Movie review dataset (Pang & Lee, 2004) for sentiment classification, Jigsaw Toxicity dataset (cjadams et al., 2017) for toxicity classification, and AGNEWS dataset (Gulli, 2005) for multi-class news categorization. The Appendix A.1 provides detailed dataset information.

**Model**   We use RoBERTa-base (Liu et al., 2019b) and BERT-base (Devlin et al., 2019) after fine-tuning on the respective datasets, and decoder-only models like LLaMA-2-7B (Touvron et al., 2023) and Qwen2.5-3B-Instruct with a task specific prompt and without finetuning. From these models, we extract paired activations: representations from a lower layer $l$ serve as input to CLVQ-VAE, while representations from a higher layer $h$ serve as reconstruction targets.

**Baseline**   For comparison, we include three baselines: the clustering-based method from Yu et al. (2024), a single-layer VQ-VAE variant on layer $l$ (referred to as "Single-Layer"), and a cross-layer sparse autoencoder (SAE). Implementation details for each are provided in Appendix A.3.

**Activation Extraction** We utilized the NeuroX toolkit (Dalvi et al., 2023) to extract paired activations from the 4 evaluated models. A key advantage of NeuroX is its ability to aggregate sub-word token representations of a model tokenizer into word-level activations. This aggregation enables the mapping of discovered concepts to complete words rather than fragmented tokens, allowing us to employ visualization techniques such as word clouds to represent and analyze the discrete concepts identified by CLVQ-VAE.

**Layer-pair of analysis** For our primary evaluation across models and datasets, we apply CLVQ-VAE between intermediate-to-upper layer pairs: layers 8–12 for BERT/RoBERTa, layers 28–32 for LLaMA-2-7B, and layers 32–36 for Qwen2.5-3B-Instruct. This specific layer pair is chosen based on theoretical and empirical evidence, which we detail in Appendix D.2.

## 4 Evaluation

We evaluate the concepts discovered by CLVQ-VAE through quantitative and qualitative analyses to answer three key research questions:

- **RQ1 (Faithfulness):** Does CLVQ-VAE identify concepts that are functionally important to the model's predictions?

- **RQ2 (Interpretability):** Are the discovered concepts semantically coherent and interpretable?

- **RQ3 (Design Choice Analysis):** How do architectural decisions in CLVQ-VAE impact concept identification?

### 4.1 RQ1: Faithfulness Evaluation via Concept Ablation

We adopt the evaluation framework from Yu et al. (2024), which measures the faithfulness of discovered concepts by ablating their representations from sentence embeddings and measuring the impact on model performance for the task.

#### 4.1.1 Methodology

We evaluate the efficacy of CLVQ-VAE in identifying salient concepts through concept ablation experiments. For each sentence, we first identify the most salient token, then determine which codebook vector it maps to. We remove this concept vector from the sentence representation embedding for that sentence at layer $l$ and measure the resulting drop in probe model performance to assess concept faithfulness.

To identify the most salient token in each input sentence, we use Layer Integrated Gradients (Layer IG) (Sundararajan et al., 2017b). This attribution method quantifies the contribution of each token's embedding to the model's final prediction, and the token with the highest attribution score is considered the most salient. Our model-specific implementations are detailed in the Appendix B.2.

We then construct three variants of the sentence representation embeddings:

- **Original CLS**: Unmodified sentence representation.

- **Perturbed CLS**: Sentence representation with the most salient concept removed via orthogonal projection. [2] For each method (Clustering, Single-Layer VQ-VAE, SAE, and CLVQ-VAE), we identify the concept vector associated with the salient token. In VQ-VAE-based approaches, this corresponds to the nearest codebook vector. In clustering, it is the closest cluster centroid. For SAE, it is the decoder vector associated with the highest activated neuron.

- **Random perturbed CLS**: Sentence representation with a random direction removed via orthogonal projection. This serves as a sanity check that performance drops are due to removing meaningful concepts rather than arbitrary perturbations.

---

[2]Details on the formulation of the orthogonal projection are provided in Appendix B.3.

Table 1: Faithfulness comparison of methods across models and datasets. Lower perturbed accuracy indicates more faithful concept identification. Values in parentheses show percentage accuracy drop after perturbation. **Best** and second-best results are highlighted.

| Dataset | Method | RoBERTa | BERT | Llama | Qwen |
|---|---|---|---|---|---|
| **ERASER-Movie** | Clustering | 0.6271 (28.6%) | 0.7757 (6.0%) | **0.7779 (14.1%)** | 0.6740 (22.8%) |
| | Single-Layer | 0.0633 ± 0.0022 (92.8%) | 0.7624 ± 0.0127 (7.6%) | 0.8028 ± 0.0169 (11.3%) | 0.6142 ± 0.0080 (29.6%) |
| | SAE (2048) | 0.3829 ± 0.1141 (56.4%) | 0.8653 ± 0.0096 (-4.9%) | 0.9072 ± 0.0009 (-0.2%) | 0.8684 ± 0.0092 (0.5%) |
| | SAE (4096) | 0.5288 ± 0.3022 (39.8%) | 0.8706 ± 0.0029 (-5.6%) | 0.8978 ± 0.0069 (0.8%) | 0.8625 ± 0.0064 (1.2%) |
| | **CLVQ-VAE** | **0.0594 ± 0.0009 (93.2%)** | **0.5311 ± 0.0354 (35.6%)** | 0.7851 ± 0.0606 (13.3%) | **0.6113 ± 0.0241 (30.0%)** |
| **Jigsaw** | Clustering | **0.5628 (38.3%)** | 0.7577 (15.8%) | 0.7793 (7.5%) | 0.6352 (23.6%) |
| | Single-Layer | 0.9019 ± 0.0036 (1.1%) | 0.8079 ± 0.0077 (10.2%) | **0.7622 ± 0.0199 (9.6%)** | 0.5917 ± 0.0230 (28.8%) |
| | SAE (2048 neurons) | 0.9130 ± 0.0016 (-0.1%) | 0.8994 ± 0.0047 (0.0%) | 0.8458 ± 0.0029 (-0.4%) | 0.8296 ± 0.0043 (0.2%) |
| | SAE (4096 neurons) | 0.9122 ± 0.0018 (-0.0%) | 0.8943 ± 0.0048 (0.6%) | 0.8444 ± 0.0019 (-0.2%) | 0.8347 ± 0.0043 (-0.4%) |
| | **CLVQ-VAE** | 0.6127 ± 0.0421 (32.8%) | **0.7372 ± 0.0090 (18.0%)** | 0.7931 ± 0.0101 (5.9%) | **0.5809 ± 0.0121 (30.1%)** |
| **AGNEWS** | Clustering | 0.3875 (46.7%) | 0.6675 (10.5%) | **0.8485 (4.7%)** | 0.7542 (15.0%) |
| | Single-Layer | 0.1011 ± 0.0021 (86.1%) | 0.6711 ± 0.0500 (10.0%) | 0.8744 ± 0.0142 (1.8%) | **0.7164 ± 0.0241 (19.3%)** |
| | SAE (2048) | 0.3372 ± 0.0093 (53.6%) | 0.6620 ± 0.0038 (11.2%) | 0.8942 ± 0.0059 (-0.5%) | 0.8953 ± 0.0061 (-0.9%) |
| | SAE (4096) | 0.3345 ± 0.0099 (54.0%) | 0.7047 ± 0.0262 (5.5%) | 0.8967 ± 0.0048 (-0.8%) | 0.8911 ± 0.0050 (-0.4%) |
| | **CLVQ-VAE** | **0.0992 ± 0.0035 (86.4%)** | 0.6492 ± 0.0442 (13.0%) | 0.8758 ± 0.0028 (1.6%) | 0.7536 ± 0.0040 (15.1%) |

To evaluate these variants, we train a simple probe (a 2-layer neural network with dropout) on the original sentence representations/task labels and then evaluate it on all the three variants. The core idea is that removing important concepts should cause significant performance drops, while removing random directions should have minimal impact. Methods that produce larger drops in perturbed CLS accuracy demonstrate stronger concept identification capabilities.

For encoder-based models like BERT and RoBERTa, we use the classification token ([CLS]) embedding as the sentence representation. For decoder-only models like LLaMA and Qwen, which lack a classification token, we use the mean of token embeddings (Lin et al., 2025). Throughout the paper, we will refer sentence representations as "CLS" for notational convenience, regardless of the underlying architecture.

### 4.1.2 Results: Baseline Comparison

Table 1 shows that CLVQ-VAE achieves lowest perturbed accuracy in 7 out of 12 model-dataset combinations. The percentage drops show that CLVQ-VAE consistently identifies concepts that, when removed, substantially impair performance. Single-layer approaches achieve the second-best performance in 6 out of 12 configurations, which can be viewed as a variant of CLVQ-VAE operating on a single representation space.

SAE showed inconsistent behavior with some negative accuracy drops. We attribute this to the fact that while VQ-VAE concentrates concepts, SAE "split" them across features (Bricken et al., 2023). Since perturbed accuracy is lowest when the concept vector is more aligned with the task-specific direction, the minimal drops for SAEs suggest that ablating a single decoder vector is insufficient to remove the distributed concept, whereas significant drops for VQ-VAE confirm successful concentration.

**Implementation Note:** (1) All results are averaged across 3 random seeds.[3]. (2) No standard deviation is reported for the clustering approach, as hierarchical clustering is deterministic by nature. (3) Also, CLVQ-VAE and Single-Layer results are reported for kmeans initialization in this baseline comparison. The reasoning behind this choice is detailed in section 4.3.1

### 4.2 RQ2: Interpretability Evaluation

We supplement our quantitative analysis with qualitative evaluation to assess the semantic coherence and interpretability of discovered concepts using two methods: an LLM-as-a-judge evaluation and a human study.

---

[3]Reference baseline values for faithfulness evaluation (Original CLS and Random Perturbation accuracies across all model-dataset combinations) are provided in C.3.

Table 2: LLM-judge based evaluation of methods across models and datasets.

| Method | Mean Rating ± Std | MRR | Win Rate |
|---|---|---|---|
| CLVQ-VAE | **1.807 ± 0.327** | **0.757** | **77.8%** |
| Single-Layer | 1.729 ± 0.312 | 0.438 | 47.2% |
| Clustering | 1.578 ± 0.348 | 0.424 | 33.3% |
| SAE | 1.570 ± 0.500 | 0.465 | 41.7% |

### 4.2.1 LLM-as-a-Judge Evaluation

While faithfulness metrics quantify functional importance, they do not reveal semantic coherence. We use LLM-based evaluation to assess whether discovered concepts provide plausible explanations for model predictions.

**Methodology**  Given an input text, its prediction, and the concept representation for its most salient token, we use LLM judges to evaluate concept quality. Each judge receives: (1) the input text, (2) the model's predicted class with its semantic label (e.g., "Positive" for class 1 in sentiment analysis, "Sports" for a news category), and (3) concept representations from all methods being compared.

To reduce individual biases from a single LLM, we use an ensemble of four LLMs (GPT-4o-mini, Claude 3.5 Haiku, Gemini 2.0 Flash, and Gemini 2.0 Flash Lite), excluding the judge with the highest disagreement (measured as $1 -$ average Pearson correlation). We use stratified sampling to ensure balanced representation across prediction categories: for binary tasks, we sample equally from true/false positives/negatives; for multi-class tasks, we stratify by correctness.

We construct concept representations as follows: for each test instance, we first identify its most salient token using Layer Integrated Gradients (as described in Section 4.1.1). For all methods, we identify which concept (codebook vector, cluster centroid, or SAE neuron) this token was assigned to, then find all training tokens assigned to that concept. If the concept represents more than half of the [CLS] token representation, we present it as up to 5 exemplar sentences (5-30 words each); otherwise, we extract the 10 most frequent words. If no training tokens were assigned to the concept, the representation is empty and automatically receives a rating of 1.

**Evaluation Metrics**  We evaluate concept quality using four metrics: *mean rating* (average score across judges and instances), *mean reciprocal rank* (MRR, measuring consistent top performance), *win rate* (proportion of configurations where a method ranks first), and *Kendall's W* (inter-judge agreement, with $W \geq 0.7$ indicating strong consensus (Kendall & Babington Smith, 1939)). Complete metric definitions are in Appendix C.1.

**Results**  Table 2 summarizes LLM-judge based evaluation results. CLVQ-VAE achieves the best performance with a mean rating of $1.807 \pm 0.327$, MRR of 0.757, and win rate of 77.8%, consistently ranking first or second. Single-Layer shows competitive mean performance ($1.729 \pm 0.312$) with the lowest variance, but its lower MRR (0.438) and win rate (47.2%) indicate it less frequently produces top-ranked concepts. Both CLVQ-VAE and Single-Layer used spherical initialization for this baseline comparison.

Other two baseline methods show weaker performance. SAE achieves a mean rating of $1.570 \pm 0.500$, indicating inconsistent quality across configurations. Clustering shows similar mean performance ($1.578 \pm 0.348$) but consistently weaker quality, reflected in its low win rate (33.3%) and MRR (0.424).

We calculate the inter-judge agreement via Kendall's coefficient of concordance. The results (detailed in Appendix D.7) show strong consensus ($W = 0.782$ overall), with highest agreement for Jigsaw ($W = 0.833$) and ERASER-Movie ($W = 0.828$). This validates that observed performance differences reflect quality distinctions rather than measurement noise.

Table 3: Human evaluation results comparing CLVQ-VAE with baseline clustering approach for ERASER-Movie dataset. Higher values indicate better performance across all metrics.

| Method | Fleiss' Kappa ($\kappa$) | Avg. Confidence | Model Alignment Rate |
|---|---|---|---|
| Clustering | 0.59 | 5.981 | 54.14% |
| CLVQ-VAE | **0.864** | **8.44** | **78.20%** |

### 4.2.2 Human Evaluation

To complement our LLM-based analysis, we conducted a human evaluation study comparing CLVQ-VAE with the clustering baseline. This evaluation assesses whether discovered concepts can be effectively visualized and interpreted by humans.

**Methodology** We randomly selected 19 sentences from the ERASER-Movie review dataset, while ensuring that we have an equal representation of true positives (TP), true negatives (TN), false positives (FP), and false negatives (FN) based on the model's predictions. This balanced selection allows us to evaluate concepts underlying both correct and incorrect predictions. For each sentence, we generate word clouds using the tokens mapped to the codebook vector associated with the sentence's most salient token during training, for both CLVQ-VAE and clustering. Examples of these word clouds across all four prediction categories are in Appendix E.

14 annotators participated in the study, collectively reviewing 266 samples. Each annotator saw the word cloud for each sentence without the original sentence, model prediction, or ground truth label. This way, annotators relied solely on the visualizations to infer model behavior. For each visualization, annotators were asked to:

1. Predict the model's sentiment label by only using the information in the word cloud.

2. Rate their confidence in this prediction on a scale of 1-10 (10 is highest).

**Evaluation Metrics** : We assess the visualization quality using three metrics: *Fleiss' Kappa* (inter-annotator agreement beyond chance, ranging from -1 to 1), *average confidence* (mean annotator certainty on a 1-10 scale) and *model alignment rate* (percentage of annotator predictions matching the model's actual prediction, regardless of correctness). Full metric formulations are provided in Appendix C.1.

**Results** Table 3 presents the results of the human evaluation comparing the clustering approach (Yu et al., 2024) with our CLVQ-VAE framework. CLVQ-VAE achieved substantially higher inter-annotator agreement ($\kappa = 0.864$, "almost perfect agreement") compared to clustering ($\kappa = 0.59$, "moderate agreement"), suggesting more consistent interpretations across annotators. Annotators also reported higher confidence (8.44 vs. 5.981) when interpreting CLVQ-VAE visualizations, indicating greater conceptual clarity. Additionally, model alignment was over 24 percentage points higher (78.20% vs. 54.14%), showing that CLVQ-VAE more faithfully communicates the model's reasoning.

## 4.3 RQ3: Design Choice Analysis

We also analyse the effect of key architectural choices within the CLVQ-VAE framework on model performance or convergence.

### 4.3.1 Codebook Initialization

We evaluate three codebook initialization strategies – random, k-means, and spherical k-means – across multiple models and datasets to assess their impact on quantitative performance and qualitative concept coherence.

Table 4: Faithfulness comparison of different CLVQ-VAE initialization methods across models and dataset. **Best** results are highlighted.

| Model | Dataset | Spherical | K-means | Random |
|-------|---------|-----------|---------|--------|
| BERT | ERASER-Movie | $0.6188 \pm 0.0207$ (25.0%) | **$0.5311 \pm 0.0354$ (35.6%)** | $0.5869 \pm 0.0210$ (28.8%) |
| | Jigsaw | $0.7637 \pm 0.0631$ (15.1%) | $0.7372 \pm 0.0090$ (18.0%) | **$0.7283 \pm 0.0123$ (19.0%)** |
| | AGNEWS | **$0.6303 \pm 0.0566$** (15.5%) | $0.6492 \pm 0.0442$ (13.0%) | $0.6644 \pm 0.0395$ (10.9%) |
| RoBERTa | ERASER-Movie | $0.0598 \pm 0.0006$ (93.2%) | **$0.0594 \pm 0.0009$ (93.2%)** | $0.0603 \pm 0.0011$ (93.1%) |
| | Jigsaw | $0.7176 \pm 0.0240$ (21.3%) | **$0.6127 \pm 0.0421$ (32.8%)** | $0.7171 \pm 0.1033$ (21.4%) |
| | AGNEWS | $0.1036 \pm 0.0045$ (85.8%) | **$0.0992 \pm 0.0035$ (86.4%)** | $0.1032 \pm 0.0005$ (85.8%) |
| LLaMA-2-7b | ERASER-Movie | $0.8508 \pm 0.0326$ (6.0%) | **$0.7851 \pm 0.0606$ (13.3%)** | $0.8608 \pm 0.0251$ (4.9%) |
| | Jigsaw | **$0.7819 \pm 0.0165$ (7.2%)** | $0.7931 \pm 0.0101$ (5.9%) | $0.8066 \pm 0.0128$ (4.3%) |
| | AGNEWS | $0.8826 \pm 0.0094$ (0.8%) | **$0.8758 \pm 0.0028$ (1.6%)** | $0.8942 \pm 0.0075$ (-0.5%) |
| Qwen2.5-3B | ERASER-Movie | $0.6273 \pm 0.0099$ (28.1%) | **$0.6113 \pm 0.0241$ (30.0%)** | $0.6189 \pm 0.0295$ (29.1%) |
| | Jigsaw | **$0.5606 \pm 0.0382$ (32.6%)** | $0.5809 \pm 0.0121$ (30.1%) | $0.6096 \pm 0.0341$ (26.7%) |
| | AGNEWS | $0.6883 \pm 0.0429$ (22.4%) | $0.7536 \pm 0.0040$ (15.1%) | **$0.6525 \pm 0.0621$ (26.5%)** |

Table 5: LLM-judge based evaluation of initialization methods across models and datasets.

| Method | Mean Rating $\pm$ Std | MRR | Win Rate |
|--------|----------------------|-----|----------|
| Spherical | **$1.903 \pm 0.306$** | **0.694** | **62.5%** |
| K-means | $1.841 \pm 0.330$ | 0.667 | 58.3% |
| Random | $1.800 \pm 0.390$ | 0.472 | 25.0% |

**Quantitative Analysis (Faithfulness)** Table 4 presents faithfulness evaluation results for different codebook initializations. Default k-means achieves lowest perturbed accuracy in 7 out of 12 model-dataset combinations, often outperforming the spherical variant. This suggests that traditional Euclidean distance-based partitioning may be more effective than angular similarity for embedding space organization in this context. Both k-means variants generally outperform random initialization, which introduces additional variance due to its unstructured codebook initialization.

**Qualitative Analysis (Interpretability)** Table 5 summarizes LLM judge evaluation results for different codebook initialization. Spherical initialization demonstrates the best performance, winning 62.5% of configurations and achieving the highest mean rating with the lowest variance. K-means performs comparably in mean rating but with slightly higher variance, suggesting less consistent quality. Random initialization shows substantially weaker performance, particularly evident in its low MRR (0.472) and win rate (25.0%). We observe strong inter-judge agreement (detailed in Appendix D.7) with an overall $W = 0.793$, where 9 of 12 configurations achieve $W \geq 0.7$.

The contrast between quantitative faithfulness metrics and qualitative evaluation might be revealing a practical consideration here: while k-means identifies functionally important features affecting model decisions, spherical initialization produces concepts that better align with human interpretation. This suggests initialization choice depends on the primary goal – functional faithfulness or semantic coherence.

### 4.3.2 Codebook Size

Table 6 shows the impact of varying codebook size $K$ on performance and utilization for ERASER-Movie–Roberta configuration. Although perturbed accuracy varies only slightly across configurations, perplexity reveals meaningful trends in codebook usage. At $K = 400$, CLVQ-VAE achieves a perplexity of 139.208, hence striking a good balance between capacity and efficiency. Smaller codebooks (e.g., $K = 50$) lead to high

Table 6: Impact of codebook size (K) on perturbed CLS accuracy and codebook perplexity for ERASER-Movie on RoBERTa model

.

| Codebook Size (K) | Perturbed CLS | Perplexity (Utilization %) |
|---|---|---|
| 50 | 0.0769 | 30.121 (60.2%) |
| 100 | 0.0665 | 49.600 (49.6%) |
| 400 | 0.0606 | 139.208 (34.8%) |
| 800 | 0.1214 | 168.444 (21.1%) |
| 1200 | 0.1004 | 184.895 (15.4%) |

utilization but often force multiple concepts to share the same codebook vector, reducing interpretability. In contrast, very large codebooks (e.g., $K = 800$ or $K = 1200$) underutilize available entries and may fragment the representation space.

### 4.3.3 Other Design Choices

**Commitment Loss Weight** Following van den Oord et al. (2018), we set commitment cost $\beta = 0.1$. Our ablations (Appendix D.5) confirm this balances encoder-codebook alignment with representation flexibility: higher values ($\beta \geq 0.6$) over-constrain assignments and reduce perplexity below 82, while lower values maintain diversity but risk training instability.

**Sampling Parameters** For stochastic sampling, we use temperature $\tau = 1.0$ and top-$k = 5$. While validation perplexity varies from 207 to 220 across different temperature settings (Appendix D.6), perturbed accuracy remains stable (0.0783 to 0.0911), showing limited sensitivity to these parameters. We therefore adopt conservative values that prioritize stable training dynamics while maintaining codebook diversity.

## 5 Related Work

Traditional interpretability methods for NLP models, such as the gradient-based and perturbation-based techniques (Sundararajan et al., 2017a; Kapishnikov et al., 2021; Rajagopal et al., 2021; Zhao & Aletras, 2023), assess the input feature contributions to predictions but often fail to reveal the internal decision-making processes. Similarly, the representation analysis literature provides insights into whether a predefined concept is learned in the representation and how such knowledge is structured in the neurons of the model (Dalvi et al., 2019; Sajjad et al., 2022a; Gurnee et al., 2023). However, the need for predefined concepts and annotated data for probing limits this approach (Antverg & Belinkov, 2022). The polysemantic and superpositional nature of neurons further complicates the neuron-level interpretation (Haider et al., 2025; Elhage et al., 2022; Fan et al., 2023).

Concept-based approaches aim to address some of these limitations by interpreting the model behavior through high-level concepts that are human-understandable. Techniques like TCAV measure the model sensitivity to predefined concepts via directional derivatives in the activation space (Kim et al., 2018), though they still rely on the manually specified concepts. Recent methods move toward discovering the latent concepts directly from the internal representations, which enables a deeper and more flexible understanding of the model functionality (Ghorbani et al., 2019; Dalvi et al., 2022; Jourdan et al., 2023; Yu et al., 2024).

Researchers have utilized sparse autoencoders (SAEs) (Härle et al., 2024) to extract interpretable features from the large language models (LLMs). However, studies have highlighted the challenges in their stability and utility. For instance, SAEs trained with different random seeds on the same data can learn divergent feature sets, which indicates the sensitivity to initialization (Paulo & Belrose, 2025). Furthermore, their performance on the downstream tasks does not consistently surpass the baseline methods, which questions their practical benefits (Kantamneni et al., 2025).

The cross-layer interpretability has garnered attention, with researchers introducing sparse crosscoders to capture and understand the features across different model layers (Lindsey et al., 2024). These methods facilitate the tasks like model diffing and circuit analysis, which enables the tracking of shared and unique features across the layers and provides deeper insights into the model behavior (Minder et al., 2025; Dunefsky et al., 2024b).

While VQ-VAE have shown success in the domains like image and speech processing (van den Oord et al., 2018; Łukasz Kaiser et al., 2018; Huang et al., 2023; Guo et al., 2020; Huang & Ji, 2020; Bhardwaj et al., 2022), their application in the NLP interpretability remains underexplored. The CLVQ-VAE framework addresses this gap by integrating the transcoder-inspired objectives to map the lower-layer representations to the higher-layer ones.

## 6 Conclusion

We presented CLVQ-VAE, a framework for discovering discrete concepts across transformer layers using vector quantization to collapse redundant residual-stream features into interpretable codebook vectors. Evaluation across four language models and three datasets shows that CLVQ-VAE consistently outperforms clustering, single-layer VQ-VAE, and SAE baselines in identifying functionally important and semantically coherent concepts.

Removing CLVQ-VAE-identified concepts degrades performance substantially by up to 93.4% accuracy drop. LLM judges rank our concepts first in 77.8% of comparisons. Human annotators achieve 78% model-alignment with our visualizations versus 54% for clustering methods, with higher inter-annotator agreement. These results validate that CLVQ-VAE discovers concepts that both influence predictions and align with human understanding.

Our design choices prove essential: the adaptive residual encoder balances knowledge preservation with refinement, while the cross-attention mechanism ensures the capture of distinct, task-critical concepts. Furthermore, temperature-based top-$k$ sampling maintains codebook diversity. We also uncover a trade-off in initialization, where k-means favors functional faithfulness and spherical k-means enhances semantic coherence. Overall, by integrating discrete representation learning with cross-layer analysis, CLVQ-VAE provides a robust framework for translating opaque model mechanisms into faithful, interpretable concepts.

## 7 Limitations

Despite the effectiveness of CLVQ-VAE, we acknowledge several limitations regarding resource demands, evaluation precision, and architectural transferability.

**Resource Demands** Extracting activation pairs from multiple layers requires substantial memory, especially for larger models. K-means initialization adds further computational costs that scale with dataset size and codebook dimensions. Models exceeding 70B parameters may need further optimization.

**Evaluation Precision** Our faithfulness measurement through perturbation differentiates methods but shows limited sensitivity to hyperparameter changes like temperature, top-$k$ values and codebook dimensions (see Appendix 15 for details). Also, the LLM-based evaluation shows sensitivity to how prompts are constructed and may struggle with unusual cases, requiring careful prompt refinement to maintain evaluation consistency across different inputs.

**Architectural Transferability** The framework requires training separate models for each layer combination, with optimal pairings varying by architecture. For decoder-only models, we utilize mean-pooled embeddings as substitutes for CLS tokens, though these may encode distinct information compared to dedicated classification tokens.

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

# A    Experimental Setup

## A.1    Dataset

Table 7: The data size of each benchmark used in the evaluation: the ERASER Sentiment dataset, Jigsaw Toxicity dataset, and the AGNEWS dataset

| Benchmark | Train | Dev | Tags |
|-----------|-------|-----|------|
| ERASER | 13878 | 856 | 2 |
| JIGSAW | 9000 | 800 | 2 |
| AGNEWS | 16000 | 1200 | 4 |

## A.2    Hyperparameters

Table 8 lists all hyperparameters used in our experiments. All weights use standard PyTorch random initialization.

Table 8: Hyperparameters used across all experiments.

| Component | Value |
|-----------|-------|
| *Architecture* | |
| Codebook size | 400 |
| Commitment cost ($\beta$) | 0.1 |
| Decoder layers | 6 |
| Decoder attention heads | 8 |
| Feedforward dimension | 2048 |
| Dropout | 0.1 |
| *Quantization* | |
| Sampling method | Top-$k$ temperature sampling |
| Top-$k$ | 5 |
| Temperature ($\tau$) | 1.0 |
| EMA decay ($\gamma$) | 0.99 |
| *Encoder* | |
| $\alpha$ constraint | Adaptive, max 0.5 |
| *Training* | |
| Optimizer | Adam |
| Learning rate | 5e-3 |
| Weight decay | 1e-4 (codebook and bias excluded) |
| LR scheduler | ReduceLROnPlateau |
| Batch size | 128 |
| Max epochs | 100 (early stopping enabled) |
| Random seed | 42 |

## A.3    Baseline Implementation Details

### A.3.1    Clustering Baseline: LACOAT

We implement the Latent Concept Attribution (LACOAT) method from Yu et al. (2024), which discovers latent concepts through hierarchical clustering of contextualized representations.

**Concept Discovery.** For each word $w_i$ in the training dataset $\mathcal{D}$, we extract all contextualized representations $\vec{z}_{w_i}$ from layer $l$ using NeuroX (Dalvi et al., 2023). Following Yu et al. (2024), we filter words with frequency $< 5$ and randomly sample up to 20 contextual occurrences per word. Agglomerative hierarchical clustering is then applied using squared euclidean distance and ward's minimum-variance criterion to obtain $K = 400$ cluster centroids $\{\mathbf{c}_j\}_{j=1}^{K}$, each representing a latent concept.

**Concept Assignment.** At inference, a logistic regression classifier (ConceptMapper) maps salient token representations to their nearest cluster. The classifier is trained using cross-entropy loss with L2 regularization, the lbfgs solver, and 100 maximum iterations. We use Integrated Gradients with a zero-vector baseline and 500 approximation steps to identify salient tokens, selecting those that comprise 50% of the total attribution mass.

**Faithfulness Evaluation.** For concept ablation, we use the assigned cluster centroid $\mathbf{c}_j$ as the concept vector in our orthogonal projection framework (Appendix B.3). Note that while the original LACOAT implementation removes concepts through direct subtraction of the centroid vector, we employ orthogonal projection for more targeted concept removal.

### A.3.2 Cross-Layer Sparse Autoencoder

Following Dunefsky et al. (2024b), we implement a sparse autoencoder that learns to map representations from layer $l$ to layer $h$ through a sparse latent space.

**Architecture.** The encoder projects layer $l$ representations to a high-dimensional space, and the decoder reconstructs layer $h$:

$$\mathbf{h} = \text{ReLU}(\mathbf{W}_{\text{enc}}\mathbf{x} + \mathbf{b}_{\text{enc}}) \tag{5}$$

$$\hat{\mathbf{y}} = \mathbf{W}_{\text{dec}}\mathbf{h} + \mathbf{b}_{\text{dec}} \tag{6}$$

where $\mathbf{W}_{\text{enc}} \in \mathbb{R}^{d_{\text{hidden}} \times d}$ with $d_{\text{hidden}} \in \{4096, 12288\}$. We use untied weights ($\mathbf{W}_{\text{enc}} \neq \mathbf{W}_{\text{dec}}^T$) to reduce feature suppression (Bricken et al., 2023).

**Training.** The model minimizes the reconstruction loss with an $L_1$ penalty on activations:

$$\mathcal{L} = \|\mathbf{y} - \hat{\mathbf{y}}\|_2^2 + \lambda\|\mathbf{h}\|_1 \tag{7}$$

We set $\lambda = 1e\text{-}4$ and train using Adam (lr = $5e\text{-}3$, weight decay $1e\text{-}4$), ReduceLROnPlateau scheduling (patience=5, factor=0.5), batch size 128, and early stopping (patience=10). The decoder bias is zero-initialized.

**Concept Extraction.** For ablation, we identify the neuron with the highest activation:

$$i^* = \arg\max_i h_i \tag{8}$$

We use its corresponding decoder vector $\mathbf{d}_{i^*}$ (the $i^*$-th column of $\mathbf{W}_{\text{dec}}$) as the concept vector.

### A.3.3 Single-Layer VQ-VAE

This baseline uses identical architecture and hyperparameters as CLVQ-VAE but reconstructs layer $l$ from itself rather than mapping from layer $l$ to layer $h$, isolating the contribution of cross-layer analysis.

# B Methodological Details

## B.1 EMA Update Details

During training, we perform temperature-based top-$k$ sampling to select codebook vectors, then apply EMA updates using hard assignments. For each codebook vector $\mathbf{e}_j$:

$$N_j^{(t)} = \gamma N_j^{(t-1)} + (1 - \gamma) \sum_i \mathbb{I}[j \text{ sampled for } \mathbf{z}_e^{(i)}] \tag{9}$$

$$\mathbf{m}_j^{(t)} = \gamma \mathbf{m}_j^{(t-1)} + (1 - \gamma) \sum_i \mathbf{z}_e^{(i)} \mathbb{I}[j \text{ sampled for } \mathbf{z}_e^{(i)}] \tag{10}$$

$$\mathbf{e}_j^{(t)} = \frac{\mathbf{m}_j^{(t)}}{N_j^{(t)}} \tag{11}$$

where $\gamma = 0.99$, $N_j^{(t)}$ tracks assignment counts, $\mathbf{m}_j^{(t)}$ accumulates vectors, and $\mathbb{I}[\cdot]$ indicates whether vector $j$ was selected via stochastic top-$k$ sampling. This provides stable codebook updates with improved utilization.

## B.2 Saliency Calculation Details

Our token saliency calculations are performed using the Layer Integrated Gradients (IG) method, which attributes a model's prediction back to its initial word embeddings. This approach allows us to see which input tokens were most important. However, because encoder and decoder-only models make predictions in fundamentally different ways, our attribution strategy is tailored to each architecture.

**Encoder-based Models (BERT and RoBERTa).** For standard classification models like BERT and RoBERTa, the attribution process is straightforward. These models produce a final logit score for each class. We apply Layer IG to explain the logit of the predicted class, tracing its value back to the input embeddings. This directly measures how much each token contributed to the final classification decision.

**Decoder-only Models (LLaMA and Qwen).** Decoder-only models are generative and perform next-token prediction. To adapt them for classification, we frame the task as having the model generate a single token representing the class label (e.g., "0" or "1") immediately following the input prompt. The saliency calculation, therefore, aims to explain why the model generated that specific class token. We target the logit of the predicted class token and attribute its value back to the embeddings of the original prompt. This reveals which parts of the input text were most responsible for steering the model's generation towards the final class label.

## B.3 Orthogonal Projection Details

In our faithfulness evaluation, we remove concepts from sentence representations using orthogonal projection. Given a sentence representation $\mathbf{x} \in \mathbb{R}^d$ and a concept vector $\mathbf{z}_c \in \mathbb{R}^d$, we compute the perturbed representation as:

$$\mathbf{x}_{\text{perturbed}} = \mathbf{x} - \text{proj}_{\mathbf{z}_c}(\mathbf{x}) = \mathbf{x} - \frac{\mathbf{x} \cdot \mathbf{z}_c}{\|\mathbf{z}_c\|^2} \mathbf{z}_c \tag{12}$$

where $\text{proj}_{\mathbf{z}_c}(\mathbf{x})$ denotes the orthogonal projection of $\mathbf{x}$ onto $\mathbf{z}_c$, and $\mathbf{x} \cdot \mathbf{z}_c$ represents the dot product. The concept vector $\mathbf{z}_c$ corresponds to a codebook vector for VQ-VAE-based methods, a cluster centroid for the clustering baseline, or a decoder vector for SAE.

## C  Evaluation Framework

### C.1  Evaluation Metric Formulations

#### C.1.1  LLM-as-a-Judge Metrics

1. **Mean Rating.** For each method $m$, the mean rating is computed as:

$$\text{MeanRating}(m) = \frac{1}{|C| \cdot |I| \cdot |J|} \sum_{c \in C} \sum_{i \in I} \sum_{j \in J} r_{m,c,i,j} \qquad (13)$$

where $C$ is the set of configurations (dataset-architecture pairs), $I$ is the set of instances, $J$ is the set of judges, and $r_{m,c,i,j}$ is the rating assigned by judge $j$ to method $m$ on instance $i$ in configuration $c$.

2. **Mean Reciprocal Rank (MRR).** For each configuration $c$, methods are ranked by their mean rating, with rank 1 assigned to the best-performing method. The MRR is:

$$\text{MRR}(m) = \frac{1}{|C|} \sum_{c \in C} \frac{1}{\text{rank}_{m,c}} \qquad (14)$$

where $\text{rank}_{m,c}$ is the rank of method $m$ in configuration $c$. Higher MRR indicates more consistent top performance across diverse settings.

3. **Win Rate.** The proportion of pairwise comparisons where a method achieves a higher mean rating than its competitor:

$$\text{WinRate}(m) = \frac{1}{|C| \cdot (|M| - 1)} \sum_{c \in C} \sum_{m' \neq m} \mathbb{1}[\text{MeanRating}_{m,c} > \text{MeanRating}_{m',c}] \times 100\% \qquad (15)$$

where $C$ is the set of configurations, $M$ is the set of methods, $m' \neq m$ denotes all methods except $m$, and $\text{MeanRating}_{m,c}$ is the average rating for method $m$ in configuration $c$.

4. **Kendall's W.** Kendall's coefficient of concordance measures agreement among judges on ranking methods. Given $n$ judges ranking $m$ methods:

$$W = \frac{12S}{n^2(m^3 - m)} \qquad (16)$$

where $S = \sum_{i=1}^{m} \left( R_i - \frac{n(m+1)}{2} \right)^2$ and $R_i = \sum_{j=1}^{n} r_{ij}$ is the sum of ranks assigned to method $i$ across all $n$ judges. Values of $W \geq 0.7$ indicate strong inter-judge consensus.

#### C.1.2  Human Evaluation Metrics

1. **Fleiss' Kappa ($\kappa$).** The measure of inter-annotator agreement beyond chance, ranging from -1 (worse than chance) to 1 (perfect agreement). We use the standard Fleiss' Kappa formula for $N$ instances, $n$ annotators, and $k$ categories.

2. **Average Confidence.** The mean confidence score reported by annotators across all instances and visualizations, measured on a scale from 1 (lowest) to 10 (highest).

3. **Model Alignment Rate.** The proportion of cases where annotator predictions match the model's actual prediction:

$$\text{Alignment}(m) = \frac{1}{|A| \cdot |I|} \sum_{a \in A} \sum_{i \in I} \mathbb{1}[\text{pred}_{a,i}^{(m)} = \text{pred}_{\text{model},i}] \times 100\% \qquad (17)$$

where $A$ is the set of annotators, $I$ is the set of instances, $\text{pred}_{a,i}^{(m)}$ is annotator $a$'s sentiment prediction for instance $i$ based on method $m$'s word cloud visualization, and $\text{pred}_{\text{model},i}$ is the model's actual prediction for that instance.

### C.2 LLM-as-a-Judge Prompt and Details

This section provides the exact prompt template used for the LLM-as-a-Judge evaluation.

#### C.2.1 Prompt Template

The following template was provided to each LLM judge. Placeholders like `{sentence}` were populated dynamically for each sample.

```
You are an expert AI and Linguistics researcher. Your task is to evaluate how well
each "Concept Representation" explains a model's prediction for a given sentence.

**Context:**
- Sentence: "{sentence}"
- Model's Prediction: The model classified this as '{prediction}'
  (Meaning: {label_meaning}).

**Your Task:**
For each "Concept Representation" below, rate how well it provides a plausible reason
for the model's prediction. A concept representation is a group of words or sentences
that together represent a meaningful concept.

**Key Question:** If a model only focused on this "Concept Representation", how well
would it support making a prediction of '{label_meaning}'?

**Important Guidelines:**
- Similar representations with significant overlap should receive the same rating -
  if two concepts contain many of the same words or convey similar meanings, they
  should be rated equally.
- Words are not inherently better than sentences - concept sentences may be more
  detailed, but focus on the final sentiment/meaning inferred from the concept rather
  than the level of detail.
- Be flexible with pattern matching - as long as the overall concept or general theme
  can be identified and reasonably supports the prediction, it should be considered
  a good concept even if not perfectly precise.

{guidance_text}

**Rating Rubric:**
- 3 (Good): The concept representation shows a general connection to the predicted
  label '{label_meaning}' - even if not perfectly precise, the overall theme or
  pattern is recognizable and plausibly supportive.
- 2 (Fair): The concept representation has some connection to the prediction but may
  be broad, mixed, or only partially relevant.
- 1 (Poor): The concept representation shows little to no connection to the prediction,
  is mostly irrelevant, or clearly contradicts the expected label.

**Concept Representations to Evaluate:**
---
[This section is dynamically generated based on the configurations]

Concept from Configuration: "{config_1_name}"
- Concept Words: {words_for_config_1}

Concept from Configuration: "{config_2_name}"
```

```
- Representative Sentences:
  - "{sentence_1_for_config_2}"
  - "{sentence_2_for_config_2}"

[...additional configurations...]
---

**Output Instructions:**
Respond with a single valid JSON object. Use each configuration name as a key, with
the value being an object containing:
- 'rating': An integer from 1-3 based on the rubric above
- 'reason': A brief explanation justifying your rating

Example Format:
{
  "Config_A": {
    "rating": 3,
    "reason": "Contains words that strongly support the predicted label and form a
              coherent concept"
  },
  "Config_B": {
    "rating": 2,
    "reason": "Somewhat supports the prediction but contains mixed concepts"
  },
  "Config_C": {
    "rating": 1,
    "reason": "Does not support the prediction, contains irrelevant words"
  }
}

CRITICAL: You must respond with ONLY valid JSON in the exact format requested above.
Do not include any explanatory text before or after the JSON. Your entire response
should be parseable JSON.
```

### C.2.2   Dataset-Specific Guidance

The {guidance_text} placeholder in the prompt was populated with the following instructions depending on the dataset to provide task-specific context to the LLM judges.

**Jigsaw Toxicity Detection.**

```
**Guidance for Toxicity Detection:**
Be lenient with borderline cases since non-toxic sentences can be confused with toxic
ones. Context matters greatly - strong emotions, passionate language, or criticism
doesn't automatically mean toxicity. For 'Toxic' predictions: Look for patterns
suggesting harmful intent, but accept that detection is challenging. For 'Non-toxic'
predictions: Accept concepts suggesting civil discourse, even if emotionally charged
or critical. Sarcasm and irony can be easily misinterpreted.
```

**ERASER-Movie Sentiment Analysis.**

```
**Guidance for Sentiment Analysis:**
Movie reviews are often nuanced and mixed. For positive predictions: Accept concepts
suggesting overall appreciation, enjoyment, or recommendation, even if some criticisms
are present. For negative predictions: Accept concepts suggesting overall disappointment
```

or criticism, even if some positive aspects are mentioned. Focus on the dominant sentiment direction rather than requiring pure positive/negative language.

**AGNews Topic Classification.**

```
**Guidance for Topic Classification:**
News topics frequently overlap – a tech company's earnings (Business + Science/Tech),
sports business deals (Sports + Business), or international conflicts affecting markets
(World + Business) are common. Accept concepts that show connection to the predicted
category even if they could reasonably fit multiple categories. Look for: World News
(countries, politics, conflicts, international themes), Sports (teams, games, players,
athletic activities), Business (companies, markets, financial concepts), Science/Tech
(technology, research, innovations, technical concepts).
```

### C.3 Reference Baseline Values

Table 9 provides the original CLS and random perturbed CLS accuracy values used as baselines across all faithfulness evaluation experiments. These values serve as reference points for calculating performance drops when salient concepts are removed from sentence representations.

Table 9: Reference baseline values for faithfulness evaluation across different model-dataset combinations and layer pairs.

| Model | Dataset | Layer Pair | Original CLS | Random Perturbed CLS |
|---|---|---|---|---|
| RoBERTa | ERASER-Movie | 8–12 | 0.8777 | 0.8190 |
| RoBERTa | Jigsaw | 8–12 | 0.9121 | 0.9121 |
| RoBERTa | AGNews | 8–12 | 0.7275 | 0.6875 |
| BERT | ERASER-Movie | 8–12 | 0.8248 | 0.8237 |
| BERT | Jigsaw | 8–12 | 0.8995 | 0.8995 |
| BERT | AGNews | 8–12 | 0.7458 | 0.7433 |
| LLaMA-2-7b | ERASER-Movie | 28–32 | 0.9051 | 0.9039 |
| LLaMA-2-7b | Jigsaw | 28–32 | 0.8428 | 0.8407 |
| LLaMA-2-7b | AGNews | 28–32 | 0.8900 | 0.8900 |
| Qwen2.5-3B | ERASER-Movie | 32–36 | 0.8727 | 0.8751 |
| Qwen2.5-3B | Jigsaw | 32–36 | 0.8312 | 0.8363 |
| Qwen2.5-3B | AGNews | 32–36 | 0.8875 | 0.8883 |

## D Additional Quantitative Results

### D.1 Computational Complexity Analysis

We analyze the computational complexity of CLVQ-VAE compared to SAE during training. Let $B$ denote batch size, $L$ sequence length, $D$ model dimension, $H_{sae}$ the SAE hidden dimension, $K$ the codebook size, and $N$ the number of transformer decoder layers.

#### D.1.1 SAE Complexity

The SAE performs encoder projection, ReLU activation, and decoder projection:

$$\mathcal{O}_{\text{SAE}} = \mathcal{O}(B \cdot L \cdot D \cdot H_{sae}) \tag{18}$$

where $H_{sae} \in \{4096, 12288\}$ must be substantially larger than $D$ for feature disentanglement. Complexity scales linearly with sequence length but is dominated by large matrix multiplications.

### D.1.2 CLVQ-VAE Complexity

CLVQ-VAE consists of encoder transformation, vector quantization, and transformer decoder:

$$\mathcal{O}_{\text{enc+quant}} = \mathcal{O}(B \cdot L \cdot D^2) + \mathcal{O}(B \cdot L \cdot D \cdot K) \tag{19}$$

$$\mathcal{O}_{\text{decoder}} = \mathcal{O}(N \cdot B \cdot (L^2 \cdot D + L \cdot D^2)) \tag{20}$$

$$\mathcal{O}_{\text{total}} = \mathcal{O}(B \cdot L \cdot D \cdot K) + \mathcal{O}(N \cdot B \cdot (L^2 \cdot D + L \cdot D^2)) \tag{21}$$

Complexity scales quadratically with sequence length due to self-attention in the $N = 6$ decoder layers. Vector quantization remains efficient with compact codebook size $K = 400$.

### D.2 Layer Pair Analysis

We analyze the impact of layer pair selection on concept discovery by evaluating several combinations across RoBERTa and BERT models fine-tuned on ERASER-Movie, Jigsaw, and AGNews. Our final choice of layers 8–12 is motivated both by theoretical understanding and empirical evidence.

**Theoretical Motivation.** Transformer-based architectures such as RoBERTa and BERT are known to exhibit hierarchical processing, where lower layers capture surface-level linguistic patterns and intermediate layers encode rich semantic information (Yu et al., 2024). We hypothesize that the transformation between Layers 8 and 12 best captures the transition from semantically meaningful representations to task-specific decision-making features. While our method is layer-agnostic in design, selecting this range enables optimal interpretability.

**Empirical Validation.** We conducted systematic experiments across multiple layer pairs, comparing the impact of concept removal using our perturbation-based faithfulness metric. Table 10 presents accuracy after perturbing the [CLS] token using discovered concepts, alongside baselines using original and randomly perturbed inputs.

Table 10: Layer pair analysis showing that layers 8–12 capture the most meaningful transformations across all datasets.

| Model–Dataset | Layer Pair | Perturbed CLS | Original CLS | Random Perturbed |
|---|---|---|---|---|
| RoBERTa–ERASER | 0–4 | 0.5140 | 0.4988 | 0.5012 |
| RoBERTa–ERASER | 4–8 | 0.7069 | 0.5374 | 0.5269 |
| RoBERTa–ERASER | 8–12 | 0.0583 | 0.8777 | 0.8190 |
| RoBERTa–Jigsaw | 0–4 | 0.4962 | 0.4962 | 0.4962 |
| RoBERTa–Jigsaw | 4–8 | 0.1734 | 0.7692 | 0.7653 |
| RoBERTa–Jigsaw | 8–12 | 0.5853 | 0.9121 | 0.9121 |
| RoBERTa–AGNews | 0–4 | 0.2567 | 0.2500 | 0.2500 |
| RoBERTa–AGNews | 4–8 | 0.3575 | 0.4092 | 0.3783 |
| RoBERTa–AGNews | 8–12 | 0.0967 | 0.7275 | 0.6875 |
| BERT–ERASER | 0–4 | 0.5035 | 0.5012 | 0.5024 |
| BERT–ERASER | 4–8 | 0.7593 | 0.7642 | 0.7631 |
| BERT–ERASER | 8–12 | 0.4813 | 0.8248 | 0.8237 |
| BERT–Jigsaw | 0–4 | 0.4962 | 0.4936 | 0.4936 |
| BERT–Jigsaw | 4–8 | 0.8154 | 0.8919 | 0.8906 |
| BERT–Jigsaw | 8–12 | 0.7308 | 0.8995 | 0.8995 |
| BERT–AGNews | 0–4 | 0.2408 | 0.2500 | 0.2500 |
| BERT–AGNews | 4–8 | 0.7283 | 0.8125 | 0.8175 |
| BERT–AGNews | 8–12 | 0.7117 | 0.7458 | 0.7433 |

These results support three key observations. First, early layers (0–4) encode minimal task-relevant concepts, as perturbing them leads to no meaningful change in prediction accuracy. Second, the 4–8 layer range shows inconsistent behavior across datasets–on ERASER-Movie, we observe an unexpected accuracy increase following concept perturbation, possibly due to the removal of noisy or redundant features, whereas on Jigsaw and AGNews, the expected performance drop suggests some level of concept relevance. Finally, the 8–12 configuration consistently reveals meaningful concepts across all datasets: perturbing this range significantly degrades model performance, indicating that it captures the most faithful and impactful transformations from semantic features to final task-specific representations.

### D.3 Cross-Attention Ablation Study

We ablate the cross-attention mechanism to assess its impact on codebook learning and concept quality. Table 11 compares CLVQ-VAE with and without cross-attention across all models and datasets. We measure two metrics: (1) **Faithfulness**: perturbed CLS accuracy after concept removal (lower values indicate concepts are more critical to model predictions), and (2) **Codebook Quality**: average pairwise cosine similarity between codebook vectors (lower values indicate more orthogonal, distinct concepts).

Table 11: Impact of cross-attention on faithfulness and codebook quality. With Res includes cross-attention; No Res removes it. Lower values indicate better performance for both metrics. Bold indicates better performance.

| Model | Dataset | Faithfulness (With vs. No Res) | Cosine Sim. (With vs. No Res) |
|---|---|---|---|
| RoBERTa | ERASER-Movie | 0.0594 vs 0.0560 | **0.751** vs 0.924 |
| RoBERTa | Jigsaw | 0.6127 vs **0.5152** | 0.575 vs **0.484** |
| RoBERTa | AGNews | **0.0992** vs 0.1067 | **0.906** vs 0.976 |
| BERT | ERASER-Movie | **0.5311** vs 0.7560 | **0.479** vs 0.760 |
| BERT | Jigsaw | **0.7372** vs 0.8752 | **0.312** vs 0.666 |
| BERT | AGNews | **0.6492** vs 0.7442 | **0.597** vs 0.839 |
| Qwen | ERASER-Movie | **0.6113** vs 0.7254 | **0.684** vs 0.690 |
| Qwen | Jigsaw | **0.5809** vs 0.5934 | **0.719** vs 0.746 |
| Qwen | AGNews | **0.7536** vs 0.8033 | **0.687** vs 0.790 |

Including cross-attention yields lower cosine similarity in 8 out of 9 configurations and lower faithfulness scores in 7 out of 9 configurations. The improved codebook quality (more orthogonal vectors) and stronger faithfulness (larger performance drops upon ablation) indicate that cross-attention enables the discrete bottleneck to learn more distinct, task-critical concepts.

### D.4 Adaptive Alpha Parameter Study

Table 12 shows the impact of different $\alpha$ strategies on training dynamics and codebook utilization across training epochs on ERASER-Movie dataset and RoBERTa model.

These results reveal that training of adaptive $\alpha$ behaves like a curriculum mechanism: it begins with low values that preserve the original input embeddings and gradually increases to allow more expressive transformations. For instance, the limited adaptive setting starts around 0.28 and converges to 0.45, achieving high perplexity from early epochs and maintaining it consistently. This facilitates effective concept discovery while optimizing for low validation loss. In contrast, fixed low $\alpha$ values such as 0.1 retain high perplexity but restrict the model's ability to adapt representations, resulting in higher loss. On the other hand, fixed high $\alpha$ values (e.g., 0.75 or 1.0) take significantly longer to reach useful perplexity levels, delaying convergence. Notably, when $\alpha = 0$, the encoder remains an identity function throughout training and becomes decoupled from decoder and quantization gradients, leading to stagnation.

We limit the adaptive $\alpha$ to do a maximum of 0.5 change. We noticed that allowing complete change of input embedding using adaptive $\alpha$ resulted in a high $\alpha$ which reduced final perplexity.

Table 12: Alpha parameter analysis showing perplexity evolution across training epochs and final validation loss.

| Alpha Strategy | Initial | Epoch 10 | Epoch 30 | Final | Best Val Loss |
|---|---|---|---|---|---|
| Adaptive (Limited) | 1.97 | 198.5 | 216 | 210.6 | 0.033 |
| Adaptive (Complete) | 1.86 | 25.54 | 50.70 | 63.21 | 0.033 |
| Fixed $\alpha$=0.0 | 238.1 | 237.3 | 239.8 | 237.2 | 0.045 |
| Fixed $\alpha$=0.1 | 180.9 | 232.3 | 238.1 | 230.8 | 0.040 |
| Fixed $\alpha$=0.4 | 1.95 | 126.9 | 160.2 | 157.2 | 0.036 |
| Fixed $\alpha$=0.75 | 1.077 | 1.883 | 27.517 | 40.106 | 0.033 |
| Fixed $\alpha$=1.0 | 1.155 | 2.397 | 15.219 | 147.470 | 0.032 |

## D.5 Commitment Weight Analysis

Table 13 shows the impact of commitment cost $\beta$ on codebook utilization across ERASER and Jigsaw datasets for RoBERTa model.

Table 13: Impact of commitment cost $\beta$ on validation perplexity across datasets.

| Commitment Cost ($\beta$) | ERASER Perplexity | Jigsaw Perplexity |
|---|---|---|
| 0.0 | 213.45 | 163.45 |
| 0.1 | 210.26 | 164.07 |
| 0.3 | 189.74 | 145.65 |
| 0.6 | 170.76 | 81.38 |
| 1.0 | 23.94 | 30.71 |

Higher $\beta$ values force stronger commitment to assigned codebook vectors, reducing perplexity but limiting concept diversity. Lower $\beta$ values allow more flexible assignments, promoting diverse concept identification. $\beta$=0.1 achieves optimal balance between concept diversity and training stability.

## D.6 Sampling Parameter Analysis

We analyze the impact of temperature and top-k parameters on codebook utilization and concept identification performance. Table 14 shows validation perplexity across different configurations, while Table 15 presents faithfulness evaluation results.

Table 14: Impact of temperature and top-k parameters on codebook utilization (validation perplexity) for ERASER-Movie on RoBERTa model. Higher perplexity indicates more diverse codebook usage.

| Temperature | Top-k | Validation Perplexity |
|---|---|---|
| 0.5 | 5 | 207.14 |
| 1.0 | 5 | 210.63 |
| 2.0 | 5 | 217.14 |
| 3.0 | 5 | 220.09 |
| 1.0 | 1 | 207.07 |
| 1.0 | 10 | 210.37 |
| 1.0 | 50 | 211.57 |
| 1.0 | 100 | 212.51 |

Increasing temperature from 0.5 to 3.0 increases validation perplexity from 207.14 to 220.09, reflecting greater exploration in codebook selection. For top-k values with $\tau = 1.0$, perplexity increases more gradually from 207.07 (k=1) to 212.51 (k=100), indicating that the exploration-exploitation balance shifts more gradually compared to temperature adjustments.

Table 15: Impact of temperature and top-k values on concept identification performance for ERASER-Movie on RoBERTa model. Despite significant differences in sampling parameters, perturbed CLS accuracies remain within a narrow range (0.0783–0.0911).

| Top-k | Temperature | Perturbed CLS Accuracy |
|-------|-------------|------------------------|
| 1 | 1.0 | 0.0911 |
| 10 | 1.0 | 0.0864 |
| 100 | 1.0 | 0.0817 |
| 400 | 1.0 | 0.0877 |
| 400 | 0.1 | 0.0806 |
| 400 | 1.0 | 0.0877 |
| 400 | 2.0 | 0.0783 |
| 400 | 4.0 | 0.0911 |

*Reference values:*
Original CLS: 0.7604     Random Perturbed CLS: 0.7264

While temperature and top-k parameters significantly affect codebook utilization (as measured by perplexity), perturbed accuracy exhibits limited sensitivity to these hyperparameters. They varied only within a narrow 0.0783–0.0911 range across all configurations, despite substantial differences in codebook usage patterns. This suggests that while different sampling strategies lead to different concept distributions, the resulting concepts remain comparably important for model predictions. We adopt conservative values ($\tau = 1.0$, k=5) to balance stable training dynamics with reasonable codebook diversity.

### D.7 Inter-Judge Agreement Analysis

To validate the reliability of our LLM-as-a-judge evaluation, we calculated Kendall's coefficient of concordance ($W$) across our ensemble of judges. Table 16 presents the agreement for the baseline comparison (corresponding to Section 4.2.1), and Table 17 presents the agreement for the initialization analysis (corresponding to Section 4.3.1).

Table 16: Inter-judge agreement on baseline rankings measured by Kendall's coefficient of concordance.

| Dataset | Kendall's W | Agreement Level |
|---------|-------------|-----------------|
| Jigsaw | 0.833 | Strong |
| ERASER-Movie | 0.828 | Strong |
| AGNews | 0.686 | Moderate |
| **Overall Average** | **0.782** | **Strong** |

Table 17: Inter-judge agreement on initialization method rankings measured by Kendall's coefficient of concordance.

| Dataset | Kendall's W | Agreement Level |
|---|---|---|
| Jigsaw | 0.910 | Strong |
| ERASER-Movie | 0.639 | Moderate |
| AGNews | 0.843 | Strong |
| **Overall Average** | **0.793** | **Strong** |

# E   Qualitative Analysis

To show how CLVQ-VAE represents concepts related to sentiment analysis, we examine examples from the ERASER-Movie review dataset. We present word clouds generated by our method for different prediction outcomes.

## E.1   False Negative Example

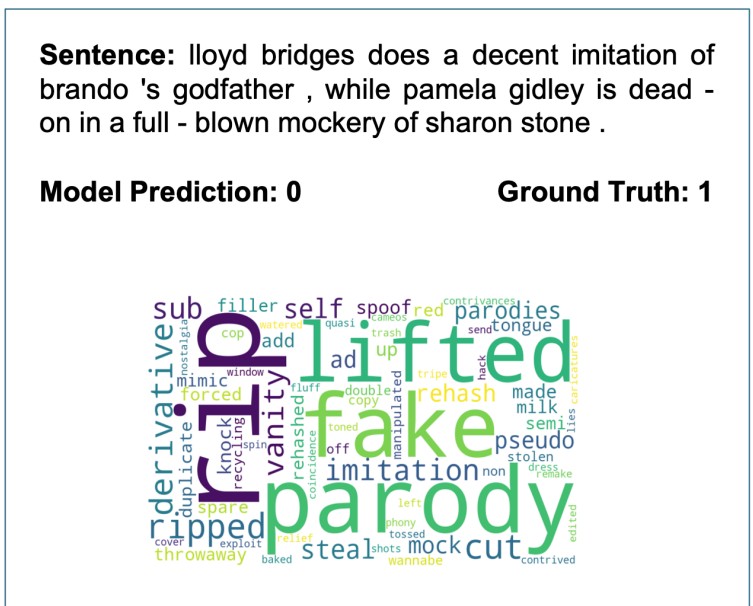

Figure 2: False negative example (Model: 0, Ground Truth: 1) showing concept clusters related to imitation and parody.

In Figure 2, we show a false negative example where the model incorrectly predicts negative sentiment for a positive review. The review describes actors performing imitations, with Lloyd Bridges doing a "decent imitation of Brando's godfather" and Pamela Gidley performing a "dead-on mockery of Sharon Stone".

The word cloud in Figure 2 contains many terms related to imitation ("lifted", "fake", "parody", "imitation", "ripped"). Despite the review framing these imitations positively as "decent" and "dead-on", the model associates these imitation concepts with negative sentiment. This shows a limitation in distinguishing between criticism of unoriginality and praise for good impersonations.

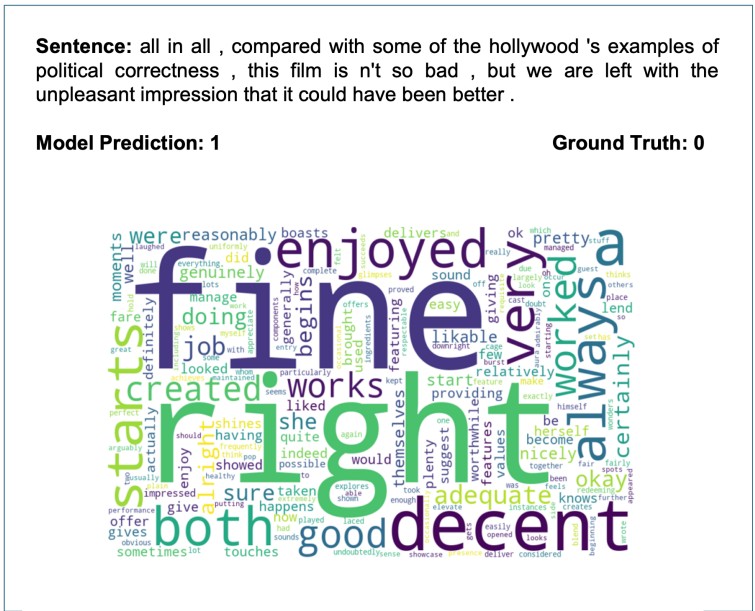

Figure 3: False positive example (Model: 1, Ground Truth: 0) showing terms of moderate approval despite the overall negative sentiment.

## E.2 False Positive Example

Figure 3 shows a false positive case where the model incorrectly predicts positive sentiment for a negative review. The review describes a film that "isn't so bad" but leaves "the unpleasant impression that it could have been better".

The word cloud in Figure 3 shows terms of moderate approval ("fine", "enjoyed", "decent", "right", "good"). The model focused on the mild praise while missing the more subtle negative sentiment. This shows a limitation in distinguishing between faint praise and genuine positive sentiment in reviews with mixed language.

## E.3 True Negative Example

Figure 4 shows a true negative example where the model correctly predicts negative sentiment. The review states "this film has neither the quality of cinematography nor the moments of glory to be highlighted".

The word cloud in Figure 4 contains terms expressing absence or lack ("nothing", "barely", "whatsoever", "little", "nowhere", "zero"). This shows how CLVQ-VAE effectively captures concepts related to deficiency, correctly identifying negative sentiment in the review.

## E.4 True Positive Example

In Figure 5, we show a true positive example where the model correctly predicts positive sentiment for "the acting is superb from everyone involved". This direct praise is an ideal case for sentiment analysis.

The word cloud in Figure 5 shows many positive descriptors ("outstanding", "fantastic", "awesome", "magnificent"). This demonstrates how our model captures related positive terms, particularly for straightforward expressions of praise.

These examples show how CLVQ-VAE captures discrete concepts for sentiment classification, which often includes sentiment-laden terms and their semantic relationships. The false prediction cases (Figures 2 and 3) highlight limitations identified by CLVQ-VAE in RoBERTa model.

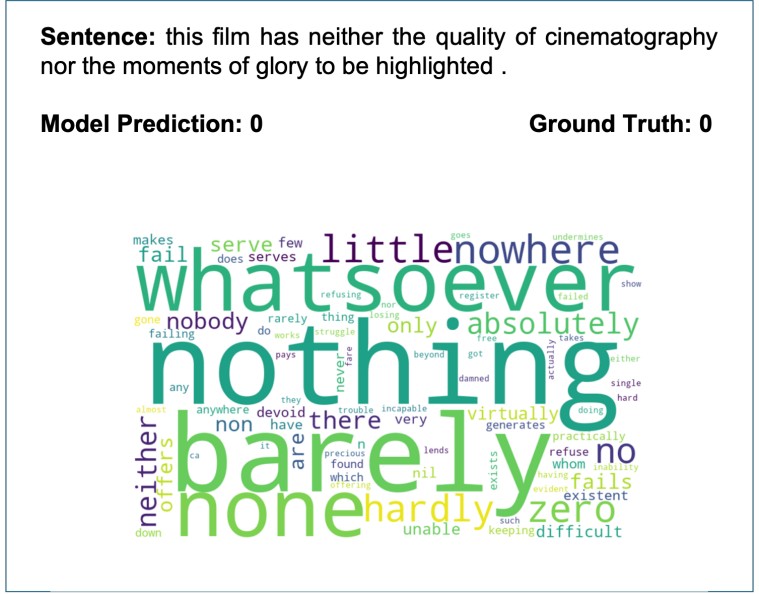

Figure 4: True negative example (Model: 0, Ground Truth: 0) showing terms expressing absence or deficiency.

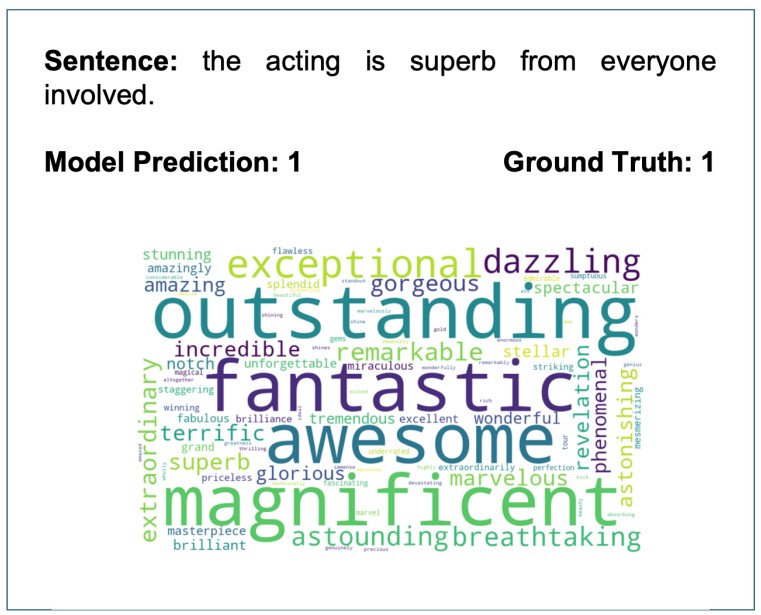

Figure 5: True positive example (Model: 1, Ground Truth: 1) showing positive descriptors for "the acting is superb from everyone involved".

