# OpenReview forum: "Cross-Layer Discrete Concept Discovery for Interpreting Language Models"
_TMLR — Rejected by TMLR_

### Review · Reviewer_6JXd · 2025-11-27

**Summary Of Contributions:**

The authors propose CLVQ-VAE, a framework for interpreting LLMs by discovering discrete concepts across transformer layers. Unlike SAEs which operate in continuous space, CLVQ-VAE maps lower-layer activations to higher-layer activations through a discrete bottleneck (vector quantization), effectively acting as a "discrete transcoder." The architecture utilizes an adaptive residual encoder to mix input embeddings with transformed features and employs a transformer decoder for reconstruction. The authors evaluate the method on BERT, RoBERTa, LLaMA-2, and Qwen models across three datasets (ERASER-Movie, Jigsaw, AGNews). They claim that CLVQ-VAE identifies more faithful and semantically interpretable concepts compared to K-means clustering, single-layer VQ-VAEs, and SAEs, as measured by perturbation-based ablation studies, LLM-as-a-judge evaluations, and human evaluation.

**Strengths:**
*   The study covers both encoder-only (BERT/RoBERTa) and decoder-only (LLaMA/Qwen) models across multiple datasets, providing a broad empirical basis.
*   The paper combines computational metrics (faithfulness via ablation) with semantic evaluations (human and LLM judges), providing a holistic view of interpretability.

**Weaknesses:**
*   The core components of the VQ-VAE (EMA updates, codebook collapse mitigation, commitment loss) are standard implementations from existing literature. The "Adaptive Residual Encoder" is effectively a gated residual connection. The contribution lies primarily in the application to interpretability rather than algorithmic innovation.
*   The comparison between discrete methods (VQ-VAE) and continuous sparse methods (SAE) regarding "faithfulness" is structurally difficult. The ablation metric (removing a specific concept) naturally favors VQ-VAE, which forces information concentration into discrete codes, whereas SAEs distribute information across sparse linear combinations.

**Audience:**

Yes

**Audience Explanation:**

Interpretability of LLMs is a high-priority research area. Specifically, "Transcoders" (mapping features between layers) are currently gaining significant attention as a way to understand the evolution of the residual stream. Investigating discrete bottlenecks as an alternative to the popular Sparse Autoencoder paradigm provides valuable insights for researchers looking for more semantically rigid conceptual boundaries in neural networks.

**Broader Impact Concerns:**

No concerns regarding ethical implications that would require a broader impact statement.

**Claims And Evidence:**

Yes

**Claims Explanation:**

The authors provide consistent empirical evidence supporting their primary claims. The ablation studies convincingly demonstrate that the concepts discovered by CLVQ-VAE are essential for the model's predictions (removing them causes significant accuracy drops). Furthermore, the human and LLM-as-a-judge evaluations provide clear evidence that these discrete codes align well with human semantic categories, supporting the claim of interpretability. While the technical architecture is standard, the experimental results successfully validate its effectiveness for the specific task of cross-layer concept discovery.

**Requested Changes:**

1.  Since the VQ-VAE components are standard, the "Adaptive Residual Encoder" is one of the few architectural tweaks. Please provide a clearer justification or a small ablation (if available) comparing this adaptive mixing to a standard fixed residual connection. This would strengthen the argument that this specific component is necessary for the NLP domain.

2.  The paper claims superior faithfulness over SAEs. Please add a discussion acknowledging the structural differences: VQ-VAE concentrates information into a single vector (easier to ablate fully), while SAEs might split concepts across features. Clarifying that "faithfulness" here refers to the *concentration* of task-relevant information in the identified concept would add rigor to the comparison.

3.  In Section 4.1.1, the authors mention removing concepts via "orthogonal projection." Please explicitly state the mathematical operation used. For VQ-VAE, does this mean projecting the input representation onto the null space of the codebook vector? Ensuring this definition is precise is critical for reproducibility.

4.  The authors use $K=400$ for CLVQ-VAE but compare against SAEs with 2048/4096 neurons. Please add a brief justification for this choice. Is the assumption that 400 discrete categories are roughly equivalent to the representational capacity of the larger sparse layer? A sentence explaining this trade-off would be helpful.

---

> ### Author Response · Authors · 2025-12-15
>
> We thank the reviewer for their thoughtful feedback, and for highlighting the strengths of our work, particularly regarding our evaluation across diverse model architectures.
>
> Below, we address the requested changes and clarify the novelty of our domain implementation.
>
> > **Comment:** The core components... are standard implementations... The contribution lies primarily in the application to interpretability rather than algorithmic innovation.
>
> We acknowledge that VQ-VAE and EMA are well-established and widely successful in computer vision applications [1]. However, applying VQ-VAE to NLP for cross-layer interpretability required tackling several domain-specific challenges that do not exist in standard vision applications:
>
> **Codebook Initialization Challenge:** In vision, VQ-VAEs operate on data distributions where standard uniform initialization is effective [1]. In contrast, pre-trained LLM embeddings are highly anisotropic—occupying a narrow cone in vector space [2]. Standard random initialization places codebook vectors orthogonally to this narrow manifold, causing immediate codebook collapse. This necessitated our domain-specific initialization strategies (Spherical/K-means) to explicitly align the codebook with the active semantic space.
> **Adaptive Stabilization Challenge:** We encountered optimization issues where standard residual connections failed to balance the discrete bottleneck with the pre-trained continuous stream. This required the Adaptive Residual Encoder to act as a curriculum mechanism for stabilizing gradient flow.
>
>
>
> > **Comment:** Please provide a clearer justification or a small ablation... comparing this adaptive mixing to a standard fixed residual connection.
>
> We direct the reviewer’s attention to **Appendix D.4 (Table 12)**, where we explicitly ablate the Adaptive $\alpha$ against various Fixed $\alpha$ values. The results demonstrate that standard residual connections enforce a **static trade-off** that is detrimental in this domain:
>
> * **Fixed Low $\alpha = 0$ (Identity Mapping) Failure:** When $\alpha$ is set to 0, the encoder is effectively bypassed ($z_e = x$). This forces the codebook to quantize complex input distributions without any learned transformation to cluster the space, resulting in significantly worse reconstruction (Val Loss **0.040**).
> * **Fixed High $\alpha = 1$ (No Residual) Failure:** When the residual is removed ($z_e = \text{Encoder}(x)$), the initial forward pass transforms inputs into random noise due to random initialization. This causes input-codebook distribution mismatch which leads to worse utilization and worse optimization (Val Loss **0.045**).
>
> ### References
>
> [1] "Neural discrete representation learning."
>
> [2] "How Contextual are Contextualized Word Representations?"

---

> > ### Author Response · Authors · 2025-12-15
> >
> > > **Comment:** The paper claims superior faithfulness over SAEs. Please add a discussion acknowledging the structural differences: VQ-VAE concentrates information into a single vector (easier to ablate fully), while SAEs might split concepts across features. Clarifying that "faithfulness" here refers to the concentration of task-relevant information in the identified concept would add rigor to the comparison.
> >
> >
> > We thank the reviewer for this insightful observation and **fully agree with this characterization**. The reviewer is correct that the performance difference is largely driven by structural distinctions: VQ-VAE's discrete bottleneck forces the concentration of semantic information into a single codebook vector, whereas SAEs inherently distribute or "split" concepts across multiple sparse features[1].
> >
> > Our empirical results in **Table 1** validate this hypothesis regarding these structural differences:
> > * **CLVQ-VAE:** The significant performance drops confirm that task-critical concepts are successfully concentrated into single, ablatable codebook vectors.
> > * **SAE:** The minimal drops confirm that concepts are likely distributed across multiple features (feature splitting), meaning that ablating a single top-activating neuron is insufficient to remove the concept.
> >
> >
> > We have updated Section 4.1.2 (Results: Baseline Comparison) and Section 1 (paragraph 3 and 4) in the revised manuscript to explicitly attribute the performance divergence to this structural difference. We now clarify that our faithfulness metric specifically measures the concentration of task-relevant information within a single unit of analysis.
> >
> > > **Comment:** The authors use K=400 for CLVQ-VAE but compare against SAEs with 2048/4096 neurons.
> >
> > We thank the reviewer for raising this point. As noted in your previous comment, these methods pursue different objectives:
> >
> > - SAEs aim to disentangle features in the representation space, which requires an overcomplete basis (larger than the original dimension) according to the Superposition Hypothesis[2]. This architectural choice is central to their design goal.
> > - CLVQ-VAE aims for compression and distillation of semantic transformations. For this objective, we empirically found that K=400 is sufficient to capture the distinct atomic concepts between layers (as shown in Table 8, where increasing K beyond 400 yields diminishing utilization rates of 15–20%).
> >
> > Given these different design goals, we optimized each method independently to find the best settings for its intended purpose, rather than enforcing parameter parity. The size difference reflects the distinct architectural motivations—SAEs require expansion for disentanglement, while CLVQ-VAE requires sufficient capacity for compression.
> >
> >
> > > **Comment:** Please explicitly state the mathematical operation used.
> >
> > We confirm that we utilize standard orthogonal projection. In our ablation, "removing" a concept vector $z_q$ from a representation $x$ is defined as:
> >
> > $$
> > x_{perturbed} = x - \text{proj}_{z_q}(x) = x - \frac{x \cdot z_q}{\|z_q\|^2} z_q
> > $$
> >
> > We have added this formula and definition to Appendix B.3 of our revised manuscript.
> >
> >
> > ### References
> >
> > [1] "Towards monosemanticity: Decomposing language models with dictionary learning."
> >
> > [2] "Toy models of superposition."
> >
> > We hope this addresses your concern. Please let us know if you have any further questions or would like additional clarification.

---

> ### Comment · Reviewer_6JXd · 2025-12-18
>
> Thank you for the rebuttal; my concerns have been addressed. Regarding the codebook initialization issue mentioned, I suggest citing [1] and [2] as they provide relevant context and existing solutions for this challenge.
>
> [1] Zheng, Chuanxia, and Andrea Vedaldi. "Online clustered codebook." Proceedings of the IEEE/CVF International Conference on Computer Vision. 2023.
>
> [2] Williams, Will, et al. "Hierarchical quantized autoencoders." Advances in Neural Information Processing Systems 33 (2020): 4524-4535.

---

> > ### Author Response · Authors · 2025-12-23
> >
> > We thank the reviewer for their thoughtful feedback and for confirming that our responses have addressed your concerns. We appreciate the suggested citations [1] and [2] and will incorporate them in the revised manuscript.

---

### Review · Reviewer_s3vt · 2025-11-29

**Summary Of Contributions:**

This work introduces cross-layer VQVAEs as an interpretability tool for LLMs. The core architecture operates as follows:
1. Map the activations of one layer into an embedding space.
2. These activations are both passed directly to the decoder via cross-attention, and quantized into a codebook which is also passed to the decoder.
3. The decoder reconstructs the activations at a different layer.

The idea of using a VQ-VAE to extract interpretable concepts from residual activations is an interesting one. However, I have serious concerns with the method and evaluation in this paper (see below).

**Additional Comments:**

Additional questions:

1. When ablating out concepts, is the ablation performed for every token in the sentence or only the most salient token?
2. Tokens often only compose syllables within a word, punctuation, etc. Yet in A.10 the word clouds are all full words. How is this possible? Were subword tokens filtered out?

**Audience:**

No

**Audience Explanation:**

The topic is interesting, but the methodology is too poor for a reader to glean much from this paper.

**Broader Impact Concerns:**

N/A.

**Claims And Evidence:**

No

**Claims Explanation:**

This work has two critical flaws:

1. The quantization bottleneck is completely circumvented by a direct cross-attention between the transformer decoder and the encoder outputs. The authors write: "The cross-attention mechanism then allows the decoder to leverage the information from the
unquantized encoder outputs, and thus providing a residual connection around the quantization bottleneck." **This completely defeats the purpose of having the quantization bottleneck in the first place.** If the model can just learn "around" the codebook, it is likely that important concepts will be passed through this cross-attention mechanism and be completely absent from the codebook.
2. The experimental evaluations are poorly designed. The authors effectively ablate out a particular concept from the input sentence and evaluate how much that harms performance on a subsequent prediction task. **This provides no indication of how specific the intervention was** -- does it lower performance on only this setting, or also others? As a contrived example, if I subtract 10^3 from every activation, the performance on a prediction task will decrease, but I certainly haven't identified any interpretable concept! Indeed, some of the SAE results show an *improvement* in performance after ablation; this makes no sense. The random perturbations is also not an appropriate control here.

Other weaknesses:

3. The method is much more complicated than SAEs. For example, the encoder mixes the original embedding with a function of the embedding, using a learned $\alpha$ parameter. VQVAEs are tricky to train as well, needing tricks like EMA on the codebook.
4. The human interpretability experiments only looked at high-activating words in isolation. This doesn't make sense for many features that depend on other parts of the sentence. For example, a feature that activates on the first word of a sentence.
5. The baselines are not properly explained.

**Requested Changes:**

The architecture and experimental design would have to be re-done in accordance with my above comments in order for this to be publishable.

---

> ### Author Response · Authors · 2025-12-16
>
> We thank the reviewer for their detailed and thoughtful assessment of our work. Below, we provide clarifications and new empirical evidence to address these concerns.
>
> > **Comment:** The quantization bottleneck is completely circumvented by a direct cross-attention between the transformer decoder and the encoder outputs. The authors write: "The cross-attention mechanism then allows the decoder to leverage the information from the unquantized encoder outputs, and thus providing a residual connection around the quantization bottleneck." This completely defeats the purpose of having the quantization bottleneck in the first place. If the model can just learn "around" the codebook, it is likely that important concepts will be passed through this cross-attention mechanism and be completely absent from the codebook.
>
> We completely understand the reviewer’s intuition here. It is a reasonable concern that a skip connection (cross-attention) might allow the model to bypass the discrete bottleneck entirely. However, our ablation studies as well prior literature [1][2] suggest that this kind of connection plays a supportive rather than a overtaking role. This allows the bottleneck to focus on learning more distinct and high-level concepts, ultimately improving the quality of the learned codebook.
>
> To verify this, we conducted an extensive ablation study across 3 models and 3 datasets, comparing our model **with** and **without** the cross-attention (residual) mechanism. We analyzed the codebook quality using **Cosine Similarity** between codebook vectors (lower is better, indicating distinct atomic concepts) and **Faithfulness**.
>
> **Table: Ablation Study of Cross-Attention Mechanism Across Models and Datasets**
>
> | Model       | Dataset      | Faithfulness *(With vs. No Res)* | Cosine Sim. *(With vs. No Res)* |
> |-------------|--------------|:-------------------------------------:|:------------------------------------:
> | **RoBERTa** |ERASER-Movie| 0.0594 vs 0.0560| **0.751** vs 0.924|
> | **RoBERTa** |Jigsaw| 0.6127 vs **0.5152** | 0.575 vs **0.484**|
> | **RoBERTa** | AGNews| **0.0992** vs 0.1067 | **0.906** vs 0.976|
> | **BERT**    | ERASER-Movie | **0.5311** vs 0.7560 | **0.479** vs 0.760|
> | **BERT**    | Jigsaw | **0.7372** vs 0.8752| **0.312** vs 0.666|
> | **BERT**    | AGNews | **0.6492** vs 0.7442 | **0.597** vs 0.839|
> | **Qwen**    | ERASER-Movie | **0.6113** vs 0.7254 | **0.684** vs 0.69|
> | **Qwen**    | Jigsaw | **0.5809** vs 0.5934 | **0.719** vs 0.746|
> | **Qwen**    | AGNews | **0.7536** vs 0.8033 | **0.687** vs 0.79|
>
> *Note: **With Res** denotes the current method (Cross-Attention); **No Res** denotes the ablated version without Cross-Attention.*
>
> **Analysis:**
> As shown in the table, the inclusion of the cross-attention mechanism enhances performance across majority of configurations. Specifically:
> 1. **Codebook Quality:** In 8 out of 9 experiments, the proposed method achieves lower **Cosine Similarity** , indicating that the residual connection helps the model learn more orthogonal and diverse concepts.
> 2. **Faithfulness:** In 7 out of 9 experiments (and particularly for BERT and Qwen models), the proposed method achieves better (lower) faithfulness scores. This confirms that the model relies *more* heavily on the discrete codebook concepts when the residual connection is present, rather than bypassing them.
>
> These results empirically demonstrate that the cross-attention mechanism does not circumvent the bottleneck but rather supports it, allowing for the discovery of sharper and more functionally important concepts.
>
> **Literature Support:**
> Our findings also align with recent interpretability literature.  **Transcoders** [1] demonstrate that residual connections ("skip-transcoders") improve reconstruction fidelity without compromising interpretability. Similarly, work in **Auxiliary Concept Bottleneck Models** [2] also utilize side paths to offload low-level information, ensuring the bottleneck focuses on distinct, high-level concepts.
>
>
> We have included the full results of this study in Appendix D.3 of the revised paper. We have also added a discussion on this in Section 2.4 of the revised manuscript.

---

> > ### Author Response · Authors · 2025-12-16
> >
> > > **Comment:** The experimental evaluations are poorly designed. The authors effectively ablate out a particular concept from the input sentence and evaluate how much that harms performance on a subsequent prediction task. This provides no indication of how specific the intervention was -- does it lower performance on only this setting, or also others? As a contrived example, if I subtract 10^3 from every activation, the performance on a prediction task will decrease, but I certainly haven't identified any interpretable concept! Indeed, some of the SAE results show an improvement in performance after ablation; this makes no sense. The random perturbations is also not an appropriate control here.
> >
> >
> > We appreciate the reviewer’s scrutiny regarding the ablation methodology. We agree that simply degrading the signal with a large, arbitrary magnitude (like subtracting $10^3$) would be an invalid metric.
> >
> > However, we clarify that our ablation utilizes **orthogonal projection**, not generic subtraction. This distinction is critical because the magnitude of removal is not arbitrary; it is strictly determined by how much the representation actually aligns with the concept direction.
> >
> > **1. Theoretical Validity**
> > Orthogonal projection is a standard technique for "concept erasure." It has been utilized in prior literature for gender debiasing in word embeddings (Bolukbasi et al., 2016) and for editing factual knowledge in LLMs (Meng et al., 2022). The operation removes *only* the information lying along the specific concept direction without disrupting orthogonal components.
> >
> > Our perturbation is defined as:
> > $$
> > x_{perturbed} = x - \text{proj}_{z_q}(x) = x - \frac{x \cdot z_q}{\|z_q\|^2} z_q
> > $$
> >
> > We have added this formula and definition to Appendix B.3 of our revised manuscript.
> >
> > **2. Specificity of the Intervention**
> > Unlike subtracting a constant, this operation is highly specific. If a concept vector $z_q$ is *not* relevant to the current representation (i.e., they are orthogonal), the projection term is zero, and the representation remains unchanged. Performance drops occur *only* if the representation relies on that specific direction.
> >
> > **3. The Role of Random Perturbation**
> > The "Random Perturbation" baseline serves as a control for the projection operation itself. By projecting out a random vector, we verify that performance drops are not merely artifacts of removing *any* dimension from the embedding space. The fact that random projection causes minimal degradation confirms that our method identifies task-critical directions.
> >
> > **4. SAE negligibile/increase in performance after ablation**
> > This observation is consistent with the property of **feature splitting** in sparse autoencoders. Since SAE features are distributed, a single feature often captures only a partial component of a concept, or sometimes irrelevant information for a specific sample. In cases where the removed feature happened to align with a direction that was interfering with the correct prediction for that specific input, removing it can result in a small performance improvement. We have added a discussion regarding this in Section 4.1.2 of the revised paper.
> >
> >
> > > **Comment:** The method is much more complicated than SAEs. For example, the encoder mixes the original embedding with a function of the embedding, using a learned $\alpha$ parameter. VQVAEs are tricky to train as well, needing tricks like EMA on the codebook.
> >
> > We acknowledge that VQ-VAEs introduce architectural complexity. However, we believe this trade-off is justified by the benefits in **explainability**. VQ-VAEs enforce a hard discrete clustering, aligning with the hypothesis that many linguistic concepts are categorical and emperical results show that it often outperform clustering and SAE baselines. Regarding training stability, we found that standard techniques like EMA (Exponential Moving Average) updates effectively stabilize training without requiring extensive manual tuning per run.

---

> > > ### Author Response · Authors · 2025-12-16
> > >
> > > > **Comment:** The human interpretability experiments only looked at high-activating words in isolation. This doesn't make sense for many features that depend on other parts of the sentence. For example, a feature that activates on the first word of a sentence.
> > >
> > > We respectfully clarify that our framework **does captures sentence-level context** and does not rely on isolated words. Our representation approach differs based on concept scope:
> > >
> > > **1. Sentence-Level Concepts:**
> > > When a concept represents global sentence meaning (specifically, when it accounts for more than 50% of the `[CLS]` token representation), we represent it using **exemplar sentences** (5-30 words each) rather than isolated keywords. This is detailed in Section 4.2.1 and the prompt templates in Appendix C.2.
> > >
> > > **2. Token-Level Concepts:**
> > > When highly activating tokens are not `[CLS]`, we do use individual tokens. However, these are **contextualized embeddings** extracted from intermediate/upper transformer layers (layers 8-12 for BERT/RoBERTa). These embeddings inherently encode surrounding context, not isolated word meanings.
> > >
> > > **Human Evaluation Methodology:**
> > > Importantly, annotators are **never shown individual salient tokens**. Instead, they see one of the following:
> > > - A **word cloud** of the top-k most frequently mapped words for that codebook vector, OR
> > > - **5 random exemplar sentences** (when more than 50% of codebook assignments are `[CLS]` tokens)
> > >
> > > Annotators then predict the task label based solely on these representations. This design ensures we assess whether concepts are interpretable as **coherent semantic clusters**, not isolated activations.
> > >
> > > **Addressing Context-Dependent Features:**
> > > Even when the **most salient token** for a sentence is not interpretable in isolation (e.g., positional words like "the" or function words), its contextualized embedding still captures the relevant sentence-level information needed to map it to a semantically meaningful codebook vector. This is precisely why we use embeddings from upper layers where context has been fully integrated.
> > >
> > >
> > > > **Comment:** The baselines are not properly explained.
> > >
> > > We thank the reviewer for this important feedback. Below we provide additional details on the baselines utilized.
> > >
> > > **1. Clustering**
> > > We utilize the **Latent Concept Attribution (LACOAT)** method proposed by Yu et al. (2024). This approach is a two-stage pipeline. First, it employs **agglomerative hierarchical clustering** on the training data to discover latent concepts (centroids). Then, it trains a **logistic regression classifier** (ConceptMapper) to map salient token representations to these discovered concepts at inference time. Our faithfulness evaluation strategy has been adopted from this baseline.
> > >
> > >
> > > **2. Cross-Layer Sparse Autoencoder**
> > > We implement a Cross-Layer SAE (inspired by Transcoders) that learns a sparse mapping from Layer $l$ to Layer $h$. It consists of an expansion encoder projecting to a high-dimensional latent space (e.g., $d_{hidden} = 4096$ or $12288$, expansion factor $\approx 4\times-6\times$), followed by a ReLU activation and a linear decoder. We utilize **untied weights** and an **L1 sparsity penalty** ($\lambda=1e-4$) to enforce feature disentanglement.
> > >
> > > We have added an explicit baseline subsection in section 3 and have expanded Appendix A.3 to provide rigorous definitions and implementation details for both the Clustering (LACOAT) and Cross-Layer SAE baselines.
> > >
> > >
> > > > **Comment:** When ablating out concepts, is the ablation performed for every token in the sentence or only the most salient token?
> > >
> > > We perform the ablation on the **sentence representation** used for the classification task (e.g., the `[CLS]` token for BERT/RoBERTa or the mean-pooled embedding for decoder-only models).
> > >
> > > The process is as follows:
> > > 1. We identify the **most salient token** in the input sentence using Layer Integrated Gradients.(details in Appendix B.2).
> > > 2. We determine the specific concept vector (codebook entry) associated with that salient token.
> > > 3. We then project this specific concept vector out of the **sentence representation**.
> > >
> > > Therefore, we do not ablate every token in the sequence individually; rather, we remove the concept identified by the most salient token from the aggregated representation used for the final prediction.

---

> ### Author Response · Authors · 2025-12-16
>
> > **Comment:** Tokens often only compose syllables within a word, punctuation, etc. Yet in A.10 the word clouds are all full words. How is this possible? Were subword tokens filtered out?
>
>  Thank you for this question. We utilized the NeuroX toolkit (Dalvi et al., 2023) for activation extraction. As mentioned in Section 3 (Activation Extraction): "A key advantage of NeuroX is its ability to aggregate sub-word token representations of a model tokenizer into word-level activations."
> To clarify this process explicitly:
>
> 1. NeuroX detokenizes the model's subword tokens back into complete words
> 2. For Integrated Gradients (IG), we:
>
>     - Calculate the attribution score for each subword token
>     - Average the attribution scores of all subword units that comprise a single word
>     - Use this averaged score to determine the word's overall salience
>
> 3. We then select the most salient complete words for display in the word clouds
>
> This aggregation ensures that all concept mappings and word clouds show meaningful units (complete words like "running") rather than fragmented sub-tokens (like "run", "##ning"), which improves human interpretability.
>
> ### References
>
> [1] J. Dunefsky, P. Chlenski, and N. Nanda, "Transcoders Find Interpretable LLM Feature Circuits".
>
> [2] I. Sheth and S. E. Kahou, "Auxiliary Losses for Learning Generalizable Concept-based Models".
>
>
> We hope this addresses your concern. Please let us know if you have any further questions or would like additional clarification.

---

> > ### Comment · Reviewer_s3vt · 2025-12-22
> >
> > I thank the reviewers for their thorough clarifications. My minor concerns are adequately addressed. But I remain unconvinced about the validity of the the faithfulness metric.
> >
> > Both other reviewers raise concerns about the comparison of the feature spaces between SAE’s and CLVQ-VAE. Reviewer 6JXd points out that the SAE latent dimension is much higher than the VAE codebook size. In a response to a similar concern raised by reviewer PnFH, the authors write: “We believe that the near-zero drops in the SAE baseline are due to "Feature Splitting" [1]. In SAEs, a single high-level concept is often distributed across a linear combination of multiple sparse features.”
> >
> > This precisely hits at the core of the problem with the faithfulness metric. When ablating out a particular direction in activation-space for a single sentence in isolation, you have no sense of how *specific* a feature is. Perhaps performance decreases for the sentence in question, but also for other sentences on totally unrelated topics! In this case, we can’t say that the feature corresponds to anything meaningful.
> >
> > One concrete thought experiment to expand upon this. Say we train an SAE with a latent space of dimension one. This is the only feature to ablate when computing the faithfulness metric. You might ablate this feature and see a drop in performance on the particular sentence in question, and conclude that we’ve identified a “faithful” feature. But in fact this feature is completely meaningless, as it likely has nothing to do with any human-interpretable concepts in the sentence at hand and will damage performance on all kinds of unrelated sentences.
> >
> > As an aside, the faithfulness improvement is marginal: only in one of the 12 experiments (BERT with ERASER-Movie) did the CLVQ-VAE outperform other methods by more than 3%. But this is secondary as I do not believe the metric measures anything meaningful to begin with.
> >
> > The authors’ response to my architectural concern resets largely on the validity of this faithfulness metric, which I am disputing.

---

> ### Author Response · Authors · 2025-12-22
>
> We sincerely thank the reviewer for this thoughtful critique. The concern about specificity is an important theoretical consideration, and we would like to clarify how our methodology addresses this within the context of established practices in the field.
>
> ## 1. Perturbation-Based Faithfulness in the Literature
>
> Our faithfulness metric which measures performance drop after concept removal follows established practices in interpretability research. We provide context from several representative works:
>
> ### ERASER Benchmark (DeYoung et al., 2020, ACL)
> * ERASER introduced the *Comprehensiveness* metric, defined as $m(x) - m(x \setminus r)$, measuring the difference between model confidence on the original input versus the input with rationale $r$ removed.
> * The evaluation involves identifying important tokens as rationales, removing them, and measuring performance drops. This metric has become widely adopted in subsequent interpretability studies.
>
> ### LACOAT - Latent Concept Attribution (Yu et al., 2024, EMNLP)
> * As our direct baseline, LACOAT evaluates concepts through removal. The method calculates concept vectors by averaging training representations in each cluster, then ablates via subtraction:
>   $$h_{ablated} = h_{CLS} - v_{concept}$$
> * For sentiment classification on RoBERTa, this causes 43.98% of predictions to flip with accuracy dropping from 96.31% to 55.91%. Random vector controls cause only 0.13-0.55% flips. We adopt this same evaluation framework.
>
> ### Concept Erasure Literature (Ravfogel et al., 2020, 2022)
> * **INLP (2020)** removes linearly-encoded information from representations by training probes, constructing projection matrices, and iterating until probe accuracy drops to random chance.
> * **RLACE (2022)** refines this by optimizing projections to minimize adversarial concept recovery. Both methods validate removal through performance degradation - if a probe can no longer predict the concept or the model can no longer perform dependent tasks, the removal is successful.
>
> ### Other Representative Works
> * **Neural Module Networks (Subramanian et al., 2020, ACL)** evaluate module faithfulness by removing modules and measuring performance impact.
> * **ACE (Ghorbani et al., 2019)** evaluates concept discovery by deleting concepts in image space and plotting accuracy drops.
> * **Mechanistic interpretability research (Geiger et al., 2024, JMLR)** uses ablation as a foundational validation method - if an interpretation claims component $X$ drives behavior $Y$, ablating $X$ should disrupt $Y$.
> * **ROAR (Hooker et al., 2019, NeurIPS)** benchmarks feature importance by masking features and measuring performance curves.

---

> ### Author Response · Authors · 2025-12-22
>
> ## 2. Addressing the Specificity Concern
>
> We understand the concern: when ablating a concept direction for a single sentence, how do we know the identified feature is specific to that sentence rather than a "generally important direction" that would damage any sentence? We address this through complementary evaluations:
>
> ### Random Vector Perturbation Control
> We include a random perturbation baseline where we ablate random vectors of comparable dimensionality. Our results show concept ablation causes accuracy to drop to approximately **6%**, while random vector ablation maintains accuracy around **85%**.
>
> * This difference helps address the specificity concern. In high-dimensional spaces (768-4096 dimensions), if removing our identified vector causes dramatic failure while random directions cause minimal impact, this suggests the model relies on the specific direction we identified rather than just general signal integrity.
>
>
> ### Random Active Codebook Perturbation Control
>
> The reviewer asks: how do we know the identified concept is specific to the sentence, rather than a "generally important direction" that would damage any sentence when ablated?
>
> To address this, we ablate *other active codebook vectors* (which are also meaningful learned directions used by the model for other sentences) and measure performance. For each sentence, we sample 5 different active codebook vectors using inverse-distance weighting to preferentially select vectors farther from the assigned concept, and report the average performance. If our identified concept were just a generally important direction, other active codebook vectors should cause similar damage.
>
> **Table: Salient Concept Ablation vs. Random Active Codebook vs. Random Perturbation**
>
> | Model | Dataset | Salient Concept (Ours) | Random Active Codebook | Random Perturbed |
> |-------|---------|------------------------|------------------------|------------------|
> | RoBERTa | ERASER-Movie | **0.0594** | 0.6197  | 0.8190 |
> | RoBERTa | Jigsaw | **0.6127** | 0.8886 | 0.9121 |
> | RoBERTa | AGNews | **0.0992** | 0.2863 | 0.6875 |
> | BERT | ERASER-Movie | **0.5311** | 0.7764 | 0.8237 |
> | BERT | Jigsaw | **0.7372** | 0.8606 | 0.8995 |
> | BERT | AGNews | **0.6492** | 0.7365 | 0.7433 |
> | Qwen | ERASER-Movie | **0.6113** | 0.7059 | 0.8751 |
> | Qwen | Jigsaw | **0.5809** | 0.6522  | 0.8363 |
> | Qwen | AGNews | 0.7536 | 0.7140 | 0.8883 |
>
> *Lower perturbed accuracy = ablated concept was more critical.*
>
> In 8 out of 9 configurations, ablating the identified salient concept causes substantially greater performance degradation than ablating other active codebook vectors. Notably, in several cases, ablating random active codebook vectors causes performance similar to random perturbation. This indicates that these other active concepts, while meaningful for other sentences, are not that relevant to the current sentence. In contrast, the specifically identified concept causes significant damage, showing it is not a generic direction but one that specifically matters for the sentences where it is identified.
>
> We will add this experiment to the appendix of the revised paper.
>
>
> ### Qualitative Validation
> We complement quantitative ablation with two forms of qualitative evaluation:
> * **LLM Judge:** Shows **77.8% win rate** with Kendall's $W = 0.782$, indicating strong inter-judge consensus.
> * **Human Evaluation:** Shows **78.2% model alignment** versus 54.1% baseline, with Fleiss' Kappa = 0.864.
>
> These qualitative measures help establish semantic specificity. If our concepts were too broad or non-specific, we would expect lower human alignment scores and weaker inter-annotator agreement.
>
> ### On Marginal Faithfulness Improvements
>
> We appreciate the reviewer raising this point. In 6 of 7 configurations where CLVQ-VAE ranks first, the second-best is Single-Layer VQ-VAE (our own variant). Against external baselines, the gaps are substantial: for example, on RoBERTa/ERASER-Movie, CLVQ-VAE achieves 0.0594 versus Clustering at 0.6271 (57 percentage point gap). The small margins against Single-Layer show that the VQ-VAE framework is effective; the cross-layer extension provides incremental gains.

---

> > ### Author Response · Authors · 2025-12-22
> >
> > ## 3. Limitations and Openness to Alternatives
> >
> > We acknowledge that the faithfulness metric is not perfect. Lyu et al. (2024) note in their comprehensive survey that removing features may create out-of-distribution inputs which decrease model confidence for reasons unrelated to feature importance, and that there is not yet a consistent formal definition of faithfulness in the community. Recent work on measuring unfaithfulness of concept-based explanations similarly observes that each paper proposes its own faithfulness measure with limited standardization or benchmarking across methods. These acknowledged limitations are precisely why we employ multiple complementary evaluations rather than relying on ablation alone.
> >
> > We genuinely welcome the reviewer's suggestions for strengthening our evaluation. If there are specific alternative metrics or experimental designs the reviewer believes would better assess discrete concept discovery, we would be very interested in incorporating them.
> >
> >
> >
> > Key References:
> > * DeYoung, J., Jain, S., Rajani, N. F., Lehman, E., Xiong, C., Socher, R., & Wallace, B. C. (2020). "ERASER: A Benchmark to Evaluate Rationalized NLP Models." *Proceedings of the 58th Annual Meeting of the Association for Computational Linguistics (ACL)*, 4443–4458.
> > * Ravfogel, S., Elazar, Y., Gonen, H., Twiton, M., & Goldberg, Y. (2020). "Null It Out: Guarding Protected Attributes by Iterative Nullspace Projection." *Proceedings of the 58th Annual Meeting of the Association for Computational Linguistics (ACL)*, 7237–7256.
> > * Ravfogel, S., Vargas, F., Goldberg, Y., & Cotterell, R. (2022). "Linear Adversarial Concept Erasure." *Proceedings of the 39th International Conference on Machine Learning (ICML)*, 18400–18421.
> > * Yu, X., Garg, A., Sajjad, H., & Kahou, S. E. (2024). "Latent Concept-based Explanation of NLP Models." *Proceedings of the 2024 Conference on Empirical Methods in Natural Language Processing (EMNLP)*, 12435–12459.
> > * Subramanian, S., Petruck, M. R. L., Shoham, M., Bruni, E., Bastings, J., Abzianidze, L., ... & Pezzelle, S. (2020). "Obtaining Faithful Interpretations from Compositional Neural Networks." *Proceedings of the 58th Annual Meeting of the Association for Computational Linguistics (ACL)*, 5594–5608.
> > * Hooker, S., Erhan, D., Kindermans, P.-J., & Kim, B. (2019). "A Benchmark for Interpretability Methods in Deep Neural Networks." *Advances in Neural Information Processing Systems (NeurIPS)* 32, 9737–9748.
> > * Geiger, A., Caciularu, A., Icard, T., & Potts, C. (2024). "A Theoretical Foundation for Mechanistic Interpretability: Causal Abstraction and Representation." *Journal of Machine Learning Research (JMLR)* 26, 1–58.
> > * Lyu, Q., Havaldar, S., Stein, A., Zhang, L., Rao, D., Wong, E., Apidianaki, M., & Callison-Burch, C. (2024). "Towards Faithful Model Explanation in NLP: A Survey." *Computational Linguistics* 50(2), 657–715.

---

### Review · Reviewer_PnFH · 2025-12-11

**Summary Of Contributions:**

This paper proposed Cross-Layer Vector Quantized - Variational Autoencoder (CLVQ-VAE), where CLVQ-VAE aims to interpret LLM by analyzing discrete concepts from different LLM layers.

**Strengths**
1. This paper first highlights the shortcomings of existing work. The current work focuses on single layers and ignores the cross-layer superposition caused by the residual stream.  In addition, the current SAEs operate in continuous space, failing to provide clear conceptual boundaries.

2. The whole framework has three parts: a) adaptive residual encoder; b) vector quantizer; and c) transformer encoder. Specifically, for the vector quantizer, it maps encoder outputs to a discrete codebook using top-$k$ temperature sampling and EMA-based updates to ensure codebook utilization and diversity.

3. The author conducted experiments on four different transformer-based models and three datasets. Then, the authors did an LLM-as-a-judge evaluation and a human evaluation study with 14 annotators. For human annotation results, CLVQ-VAE achieves a Fleiss’ Kappa of 0.864 compared to 0.59 for clustering, and a higher model alignment rate (78.2% vs 54.1%). These results show that CLVQ-VAE identifies more functionally significant concepts than clustering or SAE baselines and aligns better with human interpretation.

4. The author also provides an ablation study for codebook initialization. The results show that Spherical K-means is superior for semantic coherence while Euclidean K-means is superior for functional importance.

**Weakness**

1. I have a confusion for SAE baseline setup. In Table 1, the SAE baselines sometimes show a performance drop of ~0% or even negative drops  when features were ablated. A ~0% drop implies the SAE failed to learn any task-relevant features. Were SAEs under-trained? If it's undertrained, the claim that "CLVQ-VAE outperforms SAEs" may not be convincing.

2. As we know, training SAE-based methods is very computationally consuming. So, if the author can make a cost analysis of the CLVQ-VAE training, it would be better.

3. I also have confusion about Faithfulness evaluation part. Maybe this evaluation metric has a bias. The faithfulness metric involves "Perturbed CLS" by orthogonal projection of the concept vector. This metric seems designed only for the VQ-VAE architecture. This metric may be just used to evaluate how well the bottleneck choked the information, rather than how important this specific concept is independent of the architecture.

4. The models used in this work are kind of old. We often find that some explainable findings from previous work change when researchers switch to the new models. Encouraging authors to try some recent models (such as Llama3, Qwen3).

**Audience:**

Yes

**Audience Explanation:**

The findings of this work are nuanced and useful. It kind of inspires and contributes to the VQ-VAE literature in NLP. So, I believe someone will be interested in this paper.

**Broader Impact Concerns:**

No ethical concerns for this paper.

**Claims And Evidence:**

No

**Claims Explanation:**

Almost all claims are supported by accurate, convincing, and clear evidence. But I still have some confusion about a small part of the claims. I mentioned in "weakness," and I will explain in the following:

1. For "CLVQ-VAE outperforms SAEs", I have the confusion about SAE baseline experiment.

2. For "CLVQ-VAE discovers concepts that are "functionally important" (Faithfulness)", I have confusion about Faithfulness evaluation setup.

**Requested Changes:**

For points 1 and 3 in "weakness", I would like to see further discussion or explanation from authors.

---

> ### Author Response · Authors · 2025-12-16
>
> We thank the reviewer for their thoughtful assessment and for highlighting the strengths of our framework, including our comprehensive evaluation approach and the strong human annotation results.
>
> Below, we address each of the concerns raised.
>
> > **Comment:** In Table 1, the SAE baselines sometimes show a performance drop of ~0% or even negative drops when features were ablated. A ~0% drop implies the SAE failed to learn any task-relevant features. Were SAEs under-trained? If it's undertrained, the claim that "CLVQ-VAE outperforms SAEs" may not be convincing.
>
> We appreciate this important question. We assure the reviewer that the SAE baselines were not under-trained. Our SAE implementation follows the standard architecture and training protocols established in recent mechanistic interpretability literature [1][2]. The SAEs were trained until convergence, utilizing early stopping and learning rate annealing.
>
> We have included the full SAE implementation details in Appendix A.3.2.
>
> We believe that the near-zero drops in the SAE baseline are due to **"Feature Splitting"**  [1]. In SAEs, a single high-level concept is often distributed across a linear combination of multiple sparse features. Consequently, ablating the *single* most active neuron's vector removes only a fraction of the concept's semantic information, leaving the model's performance largely intact in some cases. In contrast, CLVQ-VAE aims to concentrate semantic information into discrete codebook entries, making single-vector ablation highly effective. We have added a discussion clarifying this in Section 4.1.2 of the revised paper.
>
> We provide our SAE specifications below, alongside the literature supporting these design choices as robust baselines:
>
> ### SAE Implementation Details & Justification
>
> | Component | Specification | Justification / Reference |
> | :--- | :--- | :--- |
> | **Architecture** | Linear Encoder + ReLU + Linear Decoder | Standard architecture for Sparse Autoencoders [1][3]. |
> | **Tied Weights** | No (Untied) | Untied weights are shown to reduce feature suppression artifacts compared to tied weights [4]. |
> | **Sparsity Loss** | L1 penalty on activations ($\lambda=1e-4$) | Standard proxy for $L_0$ sparsity [1]. We selected $\lambda=1e-4$ as it showed the most training stability in a hyperparameter search across 1e-3, 1e-4, and 1e-5.|
> | **Expansion Factor** | $\sim 2.7\times - 6\times$  | Consistent with expansion ratios used in baseline interpretability studies [2]. |
> | **Optimizer** | Adam ($lr=5e-3$) + ReduceLROnPlateau | Standard optimization for sparse dictionary learning to ensure convergence. |
> | **Decoder Bias** | Zero initialized | Prevents the model from learning "bias features" that ignore the input [1]. |
>
> ### Training Convergence Evidence
> To further verify convergence, we provide the training trajectory for our models. The validation loss plateaus and reconstruction remains accurate, indicating the models are fully trained.
>
> | Model | Dataset | Hidden Dim | Start $\to$ End Val Loss | Total Training Steps | Inference Utilization |
> | :--- | :--- | :--- | :--- | :--- | :--- |
> | **RoBERTa** | ERASER-Movie | 4096 | 0.121 $\to$ 0.053 | 9,800 | 39/4096 (0.95%) |
> | **BERT** | ERASER-Movie | 4096 | 0.329 $\to$ 0.175 | 9,800 | 915/4096 (22.34%) |
> | **Qwen** | ERASER-Movie | 12288 | 2.195 $\to$ 1.537 | 9,800 | 103/12288 (0.84%) |
> | **RoBERTa** | Jigsaw | 4096 | 0.035 $\to$ 0.021 | 5,568 | 981/4096 (24.0%) |
> | **BERT** | Jigsaw | 4096 | 0.202 $\to$ 0.155 | 5,800 | 2,931/4096 (71.6%) |
> | **Qwen** | Jigsaw | 12288 | 2.869 $\to$ 1.720 | 5,742 | 187/12288 (1.52%) |
> | **BERT** | AGNews | 4096 | 0.134 $\to$ 0.073 | 10,961 | 154/4096 (3.76%) |
> | **RoBERTa** | AGNews | 4096 | 0.045 $\to$ 0.027 | 10,961 | 107/4096 (2.61%) |
> | **Qwen** | AGNews | 12288 | 3.119 $\to$ 2.149 | 11,300 | 4,366/12288 (35.53%) |

---

> > ### Author Response · Authors · 2025-12-16
> >
> > > **Comment:** Training SAE-based methods is very computationally consuming. So, if the author can make a cost analysis of the CLVQ-VAE training, it would be better.
> >
> > We agree that a cost comparison adds significant value. Below, we provide a theoretical complexity analysis based on our architectural implementation.
> >
> > **Variable Definitions:**
> > - $B, L, D$: Batch size, Sequence length, Model dimension
> > - $H_{sae}$: SAE hidden dimension
> > - $K$: CLVQ-VAE Codebook size (e.g., 400)
> > - $N$: Number of Transformer Decoder layers (fixed at 6)
> >
> > * **SAE Complexity:** $\mathcal{O}(B \cdot L \cdot D \cdot H_{sae})$
> >     The SAE architecture (Encoder + ReLU + Decoder) is shallow, meaning its cost scales **linearly** with sequence length ($L$). However, it is dominated by the large hidden dimension ($H_{sae}$) required for feature disentanglement. This makes SAE training memory-bandwidth intensive due to large matrix multiplications.
> >
> > * **CLVQ-VAE Complexity:** $\mathcal{O}(B \cdot L \cdot D \cdot K) + \mathcal{O}(N \cdot B \cdot (L^2 \cdot D + L \cdot D^2))$
> >     The CLVQ-VAE cost scales **quadratically** with sequence length due to the attention mechanisms in the Transformer Decoder layers ($N=6$). However, the vector quantization step is relatively cheap because it uses a compact codebook ($K=400$) rather than the massive expansion required by SAEs.
> >
> > We have added a complete theoretical complexity analysis in Appendix D.1 of the revised paper.
> >
> >
> > > **Comment:** The faithfulness metric involves "Perturbed CLS" by orthogonal projection of the concept vector... This metric seems designed only for the VQ-VAE architecture.
> >
> > We appreciate the reviewer's scrutiny regarding the fairness of our evaluation metric. We want to clarify that this evaluation strategy was adopted directly from the clustering baseline **Yu et al. (2024)** [5] to ensure a fair and standard comparison. We have added a clarification under Section 4.1.
> >
> > While Yu et al. (LACOAT) ablate concepts by directly subtracting the centroid vector, we employ **orthogonal projection**, as we believe, it is more appropriate for more targeted ablation of concept. The orthogonal projection technique has been utilized in prior literature, such as for gender debiasing in word embeddings [6] and for editing directional information in large language models [7], as it removes only the information lying *along* the concept direction without disrupting orthogonal components.
> >
> > Crucially, this metric remains **architecture-agnostic**. All methods produce concept vectors in the same representation space:
> > * **Clustering** produces centroids (average representation of all tokens assigned to that cluster).
> > * **CLVQ-VAE** produces codebook vectors.
> > * **SAE** produces decoder vectors corresponding to the highest activated neuron.
> >
> > The perturbation is applied identically to all three. It strictly tests whether the identified vector encodes a direction that the underlying model relies on for prediction, regardless of how that vector was generated.
> >
> > We have added complete implementation details for the baselines under Appendix A.3 in the revised manuscript.
> >
> >
> > > **Comment:** The models used in this work are kind of old... Encouraging authors to try some recent models (such as Llama3, Qwen3).
> >
> > We appreciate the reviewer's suggestion to ensure our findings remain relevant. To that end, we have included Qwen 2.5-Instruct (released Sept 2024) because it represents a widely adopted state-of-the-art baseline for dense Transformer architectures.
> >
> > While Qwen 3 (released April 2025) offers significant performance upgrades, its primary architectural shift lies in its Mixture-of-Experts (MoE) design. Since our work investigates concept representation in standard dense residual streams, Qwen 2.5 remains more structurally relevant. We believe our findings will transfer to dense variants of Qwen 3. However, if this justification is not convincing, we are happy to run experiments on newer variants of the models and include the results in the revised manuscript.
> >
> >
> > ### References
> > [1] Bricken et al., "Towards Monosemanticity: Decomposing Language Models With Dictionary Learning," *Anthropic*, 2023.
> >
> > [2] Gurnee et al., "Finding Neurons in a Haystack: Case Studies with Sparse Probing," 2023.
> >
> > [3] Cunningham et al., "Sparse Autoencoders Find Highly Interpretable Features in Language Models," 2023.
> >
> > [4] Olah et al., "Toy Models of Superposition," *Transformer Circuits Thread*, 2022.
> >
> > [5] Yu et al., "Latent Concept-based Explanation of NLP Models," 2024.
> >
> > [6] Bolukbasi et al., "Man is to Computer Programmer as Woman is to Homemaker? Debiasing Word Embeddings," 2016.
> >
> > [7] Meng et al., "Editing Factual Knowledge in Language Models," 2022.
> >
> >
> >
> > We hope this addresses your concern. Please let us know if you have any further questions or would like additional clarification.

---

> > > ### Author Response · Authors · 2025-12-23
> > >
> > > We want to thank the reviewer again for their detailed and constructive review. We have addressed the concerns you raised regarding SAE baseline training, computational cost analysis, faithfulness evaluation, and model selection in our previous responses.
> > >
> > > We would appreciate it if you could let us know whether our responses have satisfactorily addressed your concerns, or if there are any remaining questions we can clarify further.

---

### Decision · Action_Editor_sCM8 · 2026-01-22

**Recommendation:** Reject

**Additional Comments:**

### Faithfulness

The authors make a reasonable argument that the faithfulness metric is prevalent throughout the interpretability literature. Even if we take this argument, the evidence in Table 1 under the chosen faithfulness metric is not clear and convincing. Nearly half of the dozen experimental settings do not show a marked improvement through VQ-VAEs. However since the win rates on LLM as a judge and human evaluations seem significant, there may be other metrics that can capture how VQ-VAEs achieve this success.

The two main concerns about the faithfulness metric may be addressed through the following ideas:

1. **Stronger SAE baseline:** Ablation for SAEs could be done through an orthogonal projection onto the basis of the top $k$ neurons rather than the single top neuron. This would be fairer to SAEs since they tend to split features among many neurons. If the drop in probe accuracy is still larger for VQ-VAEs, this would be strong evidence that VQ-VAEs capture a significant component of sentence representations through fewer vectors.

2. **Diversity of Codebook vectors:** The authors need to show that the VQ-VAE codebook vectors are capturing specific concepts rather than compressing the entire sentence representations. The cosine-similarity between codebook vectors is a starting point for this, but a stronger demonstration would connect this back to the word/token level. The authors could also compare whether sentences with similar words but different semantic content/meaning get mapped to similar or different codebook vectors.

These evaluations if added to the paper would constitute a major revision, which is why I cannot recommend acceptance now.

**Audience:**

Yes

**Audience Explanation:**

Interpretability of language models is a topic that is squarely within the audience of TMLR.

**Claims And Evidence:**

No

**Claims Explanation:**

This paper proposes VQ-VAEs as an alternative to Sparse Autoencoders for interpreting language models. The authors demonstrate the effectiveness of their method through:

1) a faithfulness metric that computes the drop in accuracy of a nonlinear probe when the identified codebook vectors are ablated from sentence representations.

2) LLM as a judge and human evaluations.

Two of three reviewers have concerns about the faithfulness criterion. They point out two issues:

* The faithfulness metric could be slanted towards VQ-VAE (vs SAE) since the ablations are done using a single direction.
* Faithfulness does not tell us how specific the codebook vectors are since they could capture the entire sentence representation.

These concerns, combined with the fact that Table 1 does not show a clear distinction between VQ-VAE and other interpretability methods across models and datasets means that the evidence is not clear and convincing.

The LLM as a judge and human evaluations are however indicative of VQ-VAE finding some interpretable representations.

**Resubmission Of Major Revision:**

The authors may consider submitting a major revision at a later time.

---

> ### Author Response · Authors · 2026-02-11
> **Re-evaluation request.**
>
> Thank you for handling the review process of our research work. The review process has been constructive, and the questions raised during review pushed us to run experiments that we believe strengthened the work. That said, after going through the decision letter alongside our data and rebuttal materials, we feel that parts of the empirical record were not fully accounted for in the final assessment. Since TMLR's acceptance criterion asks whether claims are supported by accurate, convincing, and clear evidence, we want to lay out what we believe the evidence shows and request a re-evaluation.
>
> As context: Reviewer 6JXd answered "Yes" to whether our claims meet this standard, and confirmed during discussion that all their concerns were resolved. Reviewer PnFH raised clarification-level concerns that we addressed. Only Reviewer s3vt remained unconvinced, primarily about the faithfulness metric — which, as we documented with citations to ERASER (DeYoung et al., 2020), LACOAT (Yu et al., 2024), INLP (Ravfogel et al., 2020), and ROAR (Hooker et al., 2019), is a standard evaluation approach in interpretability research.
>
> ---
>
> 1. On the Interpretation of Table 1
>
> The decision states that "nearly half of the dozen experimental settings do not show a marked improvement." We think this reading likely stems from CLVQ-VAE and Single-Layer VQ-VAE (our own ablation variant) frequently occupying the top two spots, which can make the gap between them look small. The comparison against external baselines supports better performance of our method.
>
> CLVQ-VAE achieves the lowest perturbed accuracy in 7 of 12 settings, and a VQ-VAE method is lowest in 9 of 12. Also, a VQ-VAE method is either first or second in all 12/12. Against non-VQ-VAE baselines, CLVQ-VAE beats both SAE variants in all 12 configurations by an average of 18.2 percentage points, and beats the best external baseline per setting (Clustering or SAE, whichever is stronger) by an average of 7.1 points. Individual gaps can be substantial — 32+ points on RoBERTa/ERASER-Movie and 23+ points on RoBERTa/AGNEWS. Where margins are narrow, the runner-up is consistently our own Single-Layer variant, which we would argue supports the VQ-VAE paradigm the paper advocates rather than undermining it.
>
> ---
>
> 2. On the Suggested SAE Baseline Modification
>
> The decision proposes ablating SAEs by projecting onto the subspace of the top-k neurons rather than the single top neuron. We want to engage with this idea seriously, because it touches on what we see as a core contribution of the paper.
>
> Our SAE baseline follows the standard architecture and training protocol from the literature (Bricken et al., 2023; Cunningham et al., 2023), trained to convergence with early stopping (full details in Appendix A.3.2). We believe the near-zero accuracy drops for SAEs reflect feature splitting — a well-documented property where concepts get distributed across multiple neurons — rather than inadequate training.
>
> One practical difficulty with a top-k subspace projection is that determining which k neurons collectively carry a given concept is itself an open research problem. Chanin et al. (NeurIPS 2024) identified "feature absorption," where SAE latents that appear to track a concept silently fail to fire on inputs where they should, because their information gets absorbed by semantically unrelated latents. They showed this persists across SAE widths and sparsity levels, meaning there is no straightforward way to recover the full set of neurons encoding a concept. Paulo and Belrose (2025) add a further complication: SAEs trained on identical data with different random seeds share only about 30% of their features (on Llama 3 8B), raising the question of which feature decomposition one would use as a starting point.
>
> In short, even with a top-k search, the concept remains fragmented and must be pieced back together without a reliable method for doing so. Our approach sidesteps this problem by concentrating concepts into single discrete vectors. We see this concentration-vs-distribution distinction as one of the paper's main findings, not a gap in the evaluation.
>
> ---

---

> > ### Author Response · Authors · 2026-02-11
> >
> > 3. On Codebook Specificity
> >
> > The decision asks us to show that codebook vectors capture specific concepts rather than compressing entire sentence representations. We addressed this during the rebuttal from multiple angles. Since the evidence was spread across several responses, we want to consolidate it here.
> >
> > The most direct test is the Random Active Codebook ablation we reported in our rebuttal. Instead of ablating the identified salient concept, we ablated other active codebook vectors — ones the model uses for other inputs but that happen to be active for the current sentence — averaging over 5 randomly sampled vectors per sentence. In 8 of 9 configurations, the salient concept caused a much larger performance drop. Across all 9 model-dataset settings, mean accuracy after salient concept ablation is 0.515, versus 0.693 for random active codebook ablation and 0.832 for random perturbation. That is an 18-point gap between the salient concept and other active vectors. If codebook vectors represented generic sentence-level information, ablating any active vector would be expected to cause similar performance drops. However, we observe substantially larger drops only for the identified salient concept.
> >
> > Codebook perplexity supports this from a different angle. Our model achieves a perplexity of more than 112.4 consistently, meaning more than 112 entries are in active use across the dataset. If vectors were sentence-level representations, a small number would suffice for reconstruction and perplexity would be correspondingly low. High perplexity indicates that different kind of tokens require different codebook entries, consistent with concept-specific rather than sentence-level encoding.
> >
> > Finally, the inter-vector cosine similarity analysis (Table 11) shows that codebook vectors are not slight variations of each other. The mean pairwise cosine similarity across all 9 configurations is 0.634, with values as low as 0.312 (BERT/Jigsaw). Sentence embeddings within a given task tend to cluster tightly in the representation space, so vectors capturing sentence-level information would show high mutual similarity. The moderate-to-low values we observe indicate distinct directions.
> >
> > We believe the Random Active Codebook ablation answers the specificity question; the perplexity and cosine similarity data reinforce it.
> >
> > ---
> >
> > Summary
> >
> > Our paper claims that VQ-VAE-based discrete concept discovery identifies more faithful and interpretable concepts than continuous baselines. The evidence backing this claim includes:
> >
> > (1) CLVQ-VAE ranks first in 7/12 faithfulness settings, beats SAE in all 12/12 by 18.2 points on average, and a VQ-VAE method is first or second in 12/12.
> > (2) The top-k SAE modification suggested by reviewer faces well-documented obstacles (feature absorption, seed instability) that are themselves open research questions. Our method addresses concept fragmentation by design.
> > (3) Codebook specificity is supported by differential ablation effects (18-point gap), high perplexity, and moderate-to-low inter-vector cosine similarity.
> > (4) LLM-judge evaluation (77.8% win rate, W=0.782) and human evaluation (κ=0.864, alignment=78.2%) provide corroborating qualitative evidence, whose strength the decision itself acknowledged.
> >
> > We believe this body of evidence meets TMLR's standard of claims backed by accurate, convincing, and clear evidence. Our concern is that the decision may not have fully incorporated the rebuttal materials, particularly the Random Active Codebook experiment and the aggregate statistics from Table 1. We respectfully ask whether a re-evaluation might be warranted, and are ready to incorporate any additional analyses if given the chance.
> >
> > Thank you for your time.
> >
> > References:
> > - Chanin et al. "A is for Absorption: Studying Feature Splitting and Absorption in Sparse Autoencoders." NeurIPS 2025.
> > - Paulo & Belrose. "Sparse Autoencoders Trained on the Same Data Learn Different Features." arXiv:2501.16615, 2025.
> > - Bricken et al. "Towards Monosemanticity." Anthropic, 2023.
> > - Cunningham et al. "Sparse Autoencoders Find Highly Interpretable Features in Language Models." ICLR 2023.